

# Mass imbalances in EPANET water-quality simulations

Michael J. Davis[1], Robert Janke[2], and Thomas N. Taxon[3]

[1]Argonne Associate of Seville, Environmental Science Division, Argonne National Laboratory, Argonne, Illinois, USA
[2]National Homeland Security Research Center, U.S. Environmental Protection Agency, Cincinnati, Ohio, USA
[3]Global Security Sciences Division, Argonne National Laboratory, Argonne, Illinois, USA

*Correspondence to:* Michael Davis (mike_davis@anl.gov) or Robert Janke (janke.robert@epa.gov)

**Abstract.** EPANET is widely employed to simulate water quality in water distribution systems. However, in general, the time-driven simulation approach used to determine concentrations of water-quality constituents provides accurate results only for short water-quality time steps. The use of an adequately short time step may not always be feasible. Overly long time steps can yield errors in concentration estimates and can result in situations in which constituent mass is not conserved. The absence of
EPANET errors or warnings does not ensure conservation of mass. This paper provides examples illustrating mass imbalances and explains how such imbalances can occur. It also presents a preliminary event-driven approach that conserves mass with a water-quality time step that is as long as the hydraulic time step. Results obtained using the current approach converge, or tend to converge, to those obtained using the preliminary event-driven approach as the water-quality time step decreases. Improving the water-quality routing algorithm used in EPANET could eliminate mass imbalances and related errors in estimated concen-
trations. The results presented in this paper should be of value to those who perform water-quality simulations using EPANET or use the results of such simulations, including utility managers and engineers.

## 1   Introduction

EPANET (Rossman, 2000; U.S. EPA, 2017a) is the standard software used for simulating water quality in a water distribution system (WDS). It has been widely and successfully applied for many years. The software includes a hydraulic model that
determines water flow and direction throughout a network model that is used to represent a WDS. The network model consists of links (pipes) and nodes (junctions). The water-quality simulation is piggybacked on the hydraulic simulation. EPANET has commonly been used in situations in which water quality does not change rapidly during the simulation. However, in some cases involving simulations of contaminant injections into a WDS it has been found that the mass of the constituent added to the network is not conserved (Davis and Janke, 2014; Davis et al., 2016). That is, at a time $t$ in a simulation, the mass of
the constituent in the network's pipes, $M_P(t)$, and tanks, $M_T(t)$, plus the cumulative mass of the constituent removed from the network by nodal demands, $MC_R(t)$, does not equal the cumulative mass of the constituent injected into the network, $MC_I(t)$. (The mass of the constituent in the system before the injection is zero and there is no loss of the constituent due to chemical reactions.) The mass imbalance can be large. For example, defining a mass-balance ratio (MBR) as $(M_P(t) + M_T(t) + MC_R(t))/MC_I(t)$, the MBR can exceed 10 or be less than 0.1 in some cases for some network models at the end
of a simulation. There can be cases in which constituent mass is gained during a simulation (MBR > 1), cases in which



constituent mass is lost (MBR < 1), as well as cases in which mass is conserved (MBR = 1). A failure to conserve constituent mass indicates that there are errors in the estimated constituent concentrations, which potentially could be a concern for any application that considers water quality in a distribution system. When poor-quality network models are used, the lack of conservation of constituent mass can be exacerbated.

EPANET is available in two forms: (1) a Microsoft Windows® version with a user interface and (2) a programmer's toolkit version. The latter consists of a dynamic link library of functions that allows software developers to customize their EPANET applications. The last major release of EPANET, including both versions, occurred in 2000 (EPANET 2.0). The last minor release of EPANET occurred in 2008 (2.00.12). In 2012, the U.S. EPA initiated a collaborative, community-based open-source effort for EPANET, the "Water Distribution Network Model" project (U.S. EPA, 2017b). In June 2015, an independent

water-community-organized, open-source project began (OpenWaterAnalytics, 2017a); an open-source-project version of the EPANET programmer's toolkit (2.1) was produced in July 2016. Also in July 2016, Lewis Rossman, the original developer of EPANET, contributed a development version (EPANET 3) of the programmer's toolkit (OpenWaterAnalytics, 2017b), which can provide mass-balance information in a status report after a water-quality simulation. None of the earlier versions of the software provides this information.

Although versions 2.00.12 and 2.1 of EPANET do not track the mass of a water-quality constituent and its location in a network during a simulation, both mass and its location can be determined using the concentrations of the constituent provided by the water-quality simulation. EPANET Example Network 3 (U.S. EPA, 2017a) is a simple network with 97 nodes; it is called Network N1 in this paper. Considering independent contaminant injections at each of the nodes in the network, Fig. 1 shows how MBRs determined for these injections are distributed after a 24 h simulation. (Details on the method used to obtain

these results are provided below.) The figure shows that sizable imbalances can occur in this network, unless quite small water-quality time steps are used. (EPANET's default water-quality time step is 300 s.) These imbalances are the result of errors in the concentrations of the constituent determined by EPANET.

    This paper provides examples in which mass imbalances occur, discusses why they occur, and presents a preliminary approach to water-quality modeling, currently under development for use in EPANET, that can eliminate such imbalances. Both

the approach currently used in EPANET and the preliminary approach included in this paper use Lagrangian water-quality models: they follow individual parcels of water as they move through the network. However, the current EPANET approach uses a time-driven simulation model, while the preliminary approach presented here uses an event-driven one. The nature of time-driven and event-driven models is discussed later in this paper.

    The next section discusses the methods used in our analysis. The nature of the problems encountered when using the current

version of EPANET is then described, followed by a section that (1) presents an event-driven simulation method for water-quality routing for EPANET that eliminates these problems and (2) compares results obtained with the two methods. Finally, the major conclusions of the paper are presented, followed by some recommendations. Details on the time-driven and event-driven models are presented in appendices. Although the term "contaminant" is often used in this paper to refer to a water-quality constituent intentionally added to the water in a WDS, the results presented here apply to any water-quality constituent present

in a WDS.



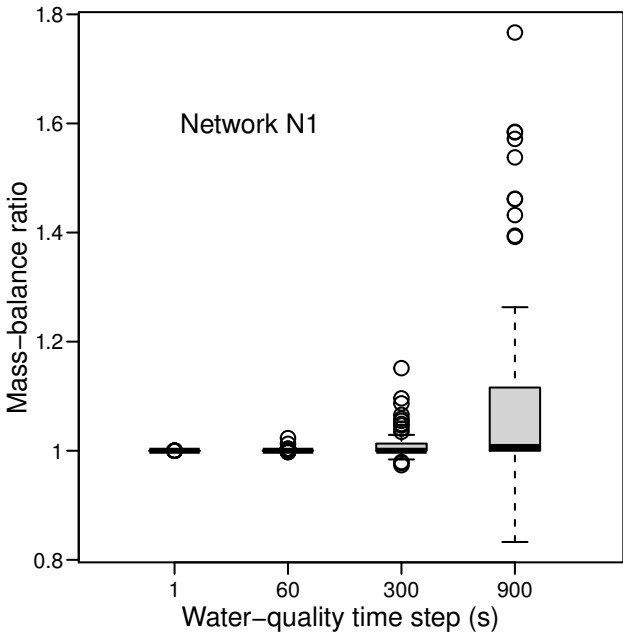

**Figure 1.** Distribution of mass-balance ratios for EPANET Example Network 3 after 24 h simulations of independent contaminant injections at each network node. The horizontal black lines in the boxplots give the median, the box extends from the lower to the upper quartile, and the whiskers extend to the smaller of 1.5 times the interquartile range or the most extreme data point. N = 94. Three nodes were excluded because there was no flow at the time the injection occurred.

## 2 Methods

The analysis for this paper was done with TEVA-SPOT (U.S. EPA, 2017c), which uses a modified version of the EPANET programmer's toolkit (2.00.12) for hydraulic and water-quality simulations in a WDS (U.S. EPA, 2017a). The version of TEVA-SPOT used was TEVA-SPOTInstaller-2.3.2-MSXb-20170110-DEV. TEVA-SPOT was developed by U.S. EPA's Na-

5      tional Homeland Security Research Center to provide an ability to evaluate the consequences of intentional and unintentional releases of a contaminant into a WDS and to design contamination warning systems for a WDS. It is the only program that we are aware of that can be used easily and efficiently to evaluate the consequences of injections at any or all nodes in a network model. The ability to track contaminant mass using the concentration results provided by EPANET was included in TEVA-SPOT to allow a better understanding of the distribution of a contaminant in a network following injection and to im-

10     prove quality control for simulations. Without this capability, the failure to conserve constituent mass that can occur during EPANET simulations would not have been identified. The only significant modification made to the EPANET 2.00.12 code to support its use in TEVA-SPOT is the inclusion of the ability to allow the direct addition of contaminant mass to tanks. Any mass imbalances identified are the result of errors in constituent concentrations provided by EPANET, not the accounting done by TEVA-SPOT.



Water-quality simulations were carried out with the time-driven water-quality model included in EPANET using four network models. Independent injection of a contaminant was simulated at all nodes in a network model and concentrations were determined at all downstream nodes for a 168 h simulation (unless noted otherwise). All simulations used 0.5 kg of contaminant injected uniformly at a rate of 8.33 g min$^{-1}$ over the period from 0:00 to 1:00 hours local time (LT), at the beginning of the

simulation (again, unless noted otherwise). In addition, the contaminant mass in pipes and tanks and the cumulative mass of contaminant withdrawn from the network were determined at each reporting step in the simulation and MBRs were calculated. Contaminants were assumed to behave as conservative tracers, with concentrations averaged over reporting intervals. Statistics on mass imbalances were determined for each network and specific injection nodes were selected for evaluation of contaminant concentrations at downstream nodes. For comparison, simulations also were done for the selected injection nodes using

the preliminary event-driven simulation model described here, with the same injection scenario as used with the time-driven method. The time and duration used for injections are arbitrary; however, a consistent injection scenario is necessary to ensure consistent hydraulic conditions for water-quality simulations.

Except as noted, all simulations used a hydraulic time step of 3600 s. Various water-quality time steps were used for time-driven simulations to determine the influence of the time step on the MBR and the constituent concentrations. The default

water-quality time step in EPANET is 300 s, as noted aboves; however, some studies, e.g, Diao et al. (2016); Helbling and VanBriesen (2009); Wang and Harrison (2014), use substantially longer time steps. Therefore, in our analysis we include water-quality time steps longer than the default value. A reporting time step of 3600 s was used in all simulations. Event-driven simulations used a water-quality time step of 3600 s. A quality tolerance of 0.01 mg L$^{-1}$ was used for all simulations, except as noted.

The network models used are summarized in Table 1. Network N1 is EPANET Example Network 3 (U.S. EPA, 2017a). Network N2 is a synthetic network called Micropolis (Brumbelow et al., 2007). Network N3 is a model for an actual distribution system that has been used in previous studies, e.g., Davis et al. (2014). Finally, Network N4 is Network 2 in the paper "The battle of the water sensor networks (BWSN)" by Ostfeld et al. (2008). The version of Network N4 used in this study is available; see the section below on code and data availability. No warnings or errors occurred while using EPANET with the network

models and cases considered in this paper. Network schematics are provided in Appendix A.

## 3   Simulations with EPANET's time-driven appoach

The time-driven approach used in EPANET is discussed and examples are provided of cases in which the approach does not conserve constituent mass.

### 3.1   Background

EPANET uses the Lagrangian time-driven simulation method for water-quality routing in a network discussed in Rossman and Boulos (1996). In general, the method may not always provide exact results. In particular, if the water-quality time step is too long, concentration errors can occur; time steps should be less than the time required for a water parcel to move through



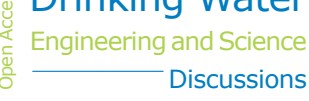

**Table 1.** Network descriptions.

| Quantity | Network | | | |
|---|---|---|---|---|
| | N1 | N2 | N3 | N4 |
| Population ($10^3$) | 79 | 5 | 130 | 250 |
| Mean water use ($m^3\,s^{-1}$) | 0.7 | 0.07 | 0.4 | 1.4 |
| Per capita use ($L\,d^{-1}$) | 760 | 1,200 | 280 | 480 |
| Nodes ($10^3$) | 0.097 | 1.6 | 6.8 | 13 |
| NZD nodes ($10^3$) | 0.059 | 0.69 | 6.7 | 11 |
| Pipes ($10^3$) | 0.12 | 1.4 | 8.0 | 15 |
| Tanks | 3 | 1 | 5 | 2 |
| Reservoirs | 2 | 2 | 1 | 2 |
| Pumps | 2 | 8 | 20 | 4 |
| Valves | 0 | 200 | 16 | 5 |

All numbers are rounded independently to two significant figures. NZD: non-zero demand.

the network pipe segment (link) having the shortest travel time for the simulation. In principle, errors can be avoided if a sufficiently short time step is used. However, such time steps may not be practical or feasible from a computational perspective. For example, pumps and values have zero length in EPANET. Also, EPANET allows a minimum time step of only 1 s, which in some cases may not be sufficiently short. Finally, water parcels can move through only one link in a water-quality time step.
The approach used in EPANET can be computationally challenging because of the possibility of a large number of links in a network and the need to use a short time step to minimize concentration errors.

    The algorithm used in EPANET to route water quality through a network can result in situations in which constituent mass is gained or lost. These imbalances can occur because of the manner in which water volume and constituent mass are accumulated at nodes and the manner in which volume and concentration are determined for subsequent releases to downstream
links. Constituent mass can be generated during the accumulation step and lost during the release step at locations for which the volume of water being moved during a water-quality time step exceeds the volume of the link in which the water is being moved. When there is a spatial gradient in constituent concentration at such locations, the mass generated during the accumulation step and lost during the release step will not be the same and a net generation or loss of constituent mass can occur. A detailed example is provided in Appendix B illustrating how the time-driven algorithm used in EPANET can fail to
conserve constituent mass in such situations.

    Restricting the movement of water parcels to only one link per water-quality time step means that, when longer time steps are used, a longer time is required for a parcel to reach a particular downstream location. For example, the arrival time of a contaminant pulse at some location following an upstream injection will be delayed if a longer time step is used. This results in





concentration errors even if the shape of the pulse in unaffected. If delays are sufficiently long, the potential exists for changes in hydraulic conditions, which could also affect concentrations. Errors in concentrations due to the effects associated with allowing water parcels to move through only one link per water-quality time step can occur even if mass is conserved.

## 3.2 Examples illustrating mass imbalances

The examples presented in this section were obtained using EPANET's time-driven water-quality routing algorithm; they demonstrate that large mass imbalances can occur, that mass-balance and concentration results can be sensitive to the water-quality time step used, and that a very short time step may be necessary to avoid significant mass imbalances and to minimize concentration errors. These results indicate that to model contaminant intrusion events more accurately a more robust algorithm is needed for use with EPANET that can ensure conservation of mass during water-quality simulations.

For a contaminant injection at a selected node in Network N3, Fig. 2 shows how the various components of mass balance change as the water-quality time step is varied using the time-driven simulation method in EPANET. Note that different vertical scales are used in each plot in the figure. The mass that is injected or removed is a *cumulative* mass; the mass in pipes and tanks is the mass in those locations at each time in the simulation. For an injection at Node 100 in Network N3, a significant mass imbalance can occur for water-quality time steps of 60 s or longer. At the end of the 168 h simulations, the MBRs are
7.11, 4.88, 1.38, and 1 for water-quality time steps of 900, 300, 60, and 1 s, respectively. For the time steps equal to or longer than 60 s, considerably more mass was removed from the network than was injected. A time step less than 60 s is necessary to conserve mass in this case.

Changes in MBRs following injections at Node IN1029 in Network N2 (Micropolis) are shown in Fig. 3 for the time-driven simulation method and four different water-quality time steps. Considerable time is required before the ratios stabilize for the
longer time steps. Only for a time step of 1 s does the MBR approximately equal 1.0. This network contains 196 valves, which, as noted, have zero length in EPANET, and, therefore, zero travel time, which likely contributes to the large MBR values shown in the figure. Note in Fig. 3 that the mass imbalances are larger for a time step of 300 s than for a time step of 900 s. The location of Node IN1029 is shown in Fig. A3.

Figure 4 shows how MBRs determined at the end of a 168 h simulation for an injection at Node 300 in Network N3 depend
on the water-quality time step used with the time-driven method. Around a time step of 30 s the mass-balance ratio has begun to diverge from 1.0, with a value of 1.06, a 6% imbalance, for that time step. A very short time step is necessary to obtain a mass-balance ratio near 1.0. Note that both scales in the figure are logarithmic.

Using two different water-quality time steps, Fig. 5 shows how the MBR varies during simulations for an injection at Node JUNCTION-3064 in Network N4 (BWSN), again using the time-driven method. Although the ratio stabilizes after about 20 h
at 1.006 for a time step of 300 s and at about 1.000 for a time step of 60 s, the ratio can be significantly different from 1.0 during the early portion of the simulations, even with a water-quality time step of 60 s. Mass is first lost from the system, then gained, then lost again, before the ratio approximately stabilizes. Note that the vertical scales on the two plots in Fig. 5 are different.

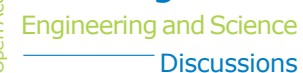



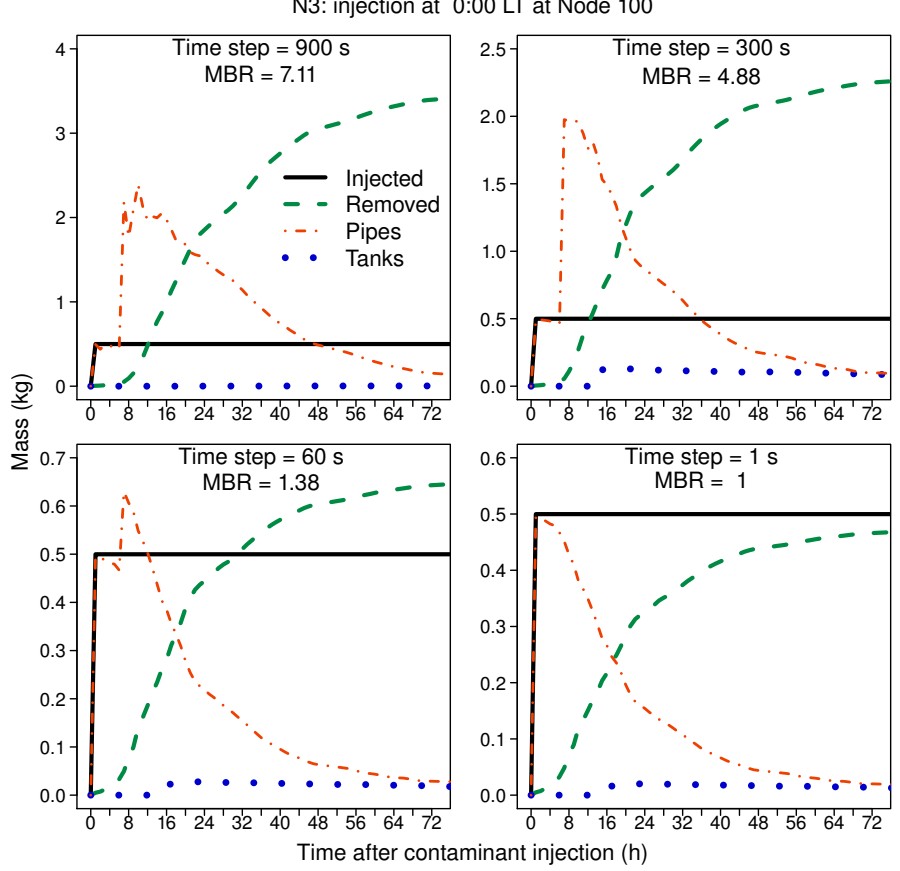

**Figure 2.** Influence of the water-quality time step on the components of mass balance for an injection at Node 100 in Network N3.

Figure 6 compares contaminant concentrations obtained using the time-driven method for Node 247 in Network N1 (EPANET Example Network 3) following an injection at Node 101, for different water-quality time steps. As the water-quality time step decreases, the magnitude and timing of the contaminant pulses at the downstream node change. The concentrations appear to stabilize when the time step is reduced to 60 s. The MBRs given in the figure are values at the end of a 24 h simulation.

5      Contaminant concentrations at Node TN1810 in Network N2 following an injection at Node IN1029 were determined using the time-driven method and are compared in Fig. 7 for different water-quality time steps. Again, the magnitude and timing of the contaminant pulses vary with the time step used. However, in this case, rather than generally decreasing as the time step decreases, the concentration of the pulse increases substantially in magnitude as the time step is reduced from 900 to 300 s, consistent with the MBRs determined for this injection node and shown in Fig. 3. The maximum concentration then decreases

10   by over 99% going from a time step of 300 s to a time step of 1 s. The MBRs given in the figure are values at the end of a 168 h simulation.

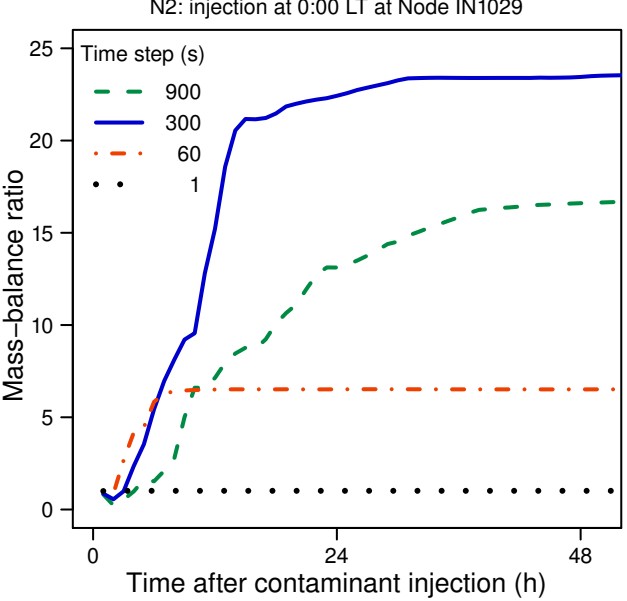

**Figure 3.** Mass-balance ratios for injections at Node IN1029 in Network N2 during simulations with different water-quality time steps. The location of Node IN1029 is shown in Fig. A3.

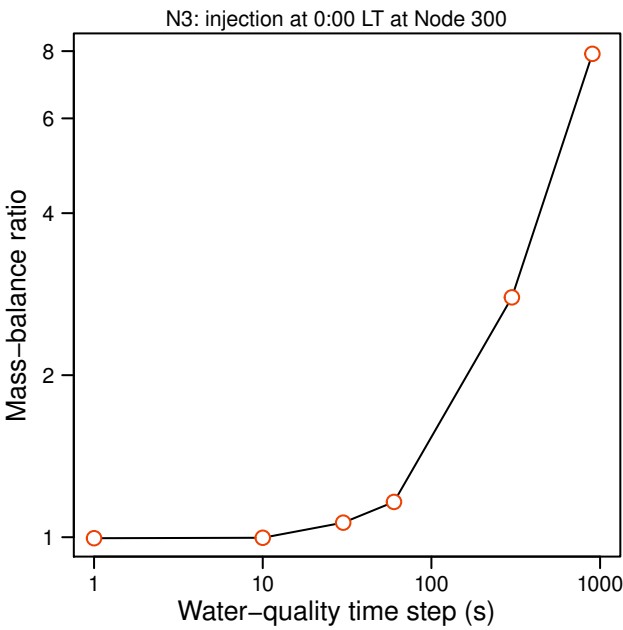

**Figure 4.** Influence of the water-quality time step on the mass-balance ratio at the end of a 168 h simulation for an injection at Node 300 in Network N3. (Note that both scales in the figure are logarithmic.)

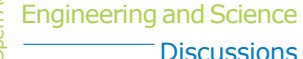

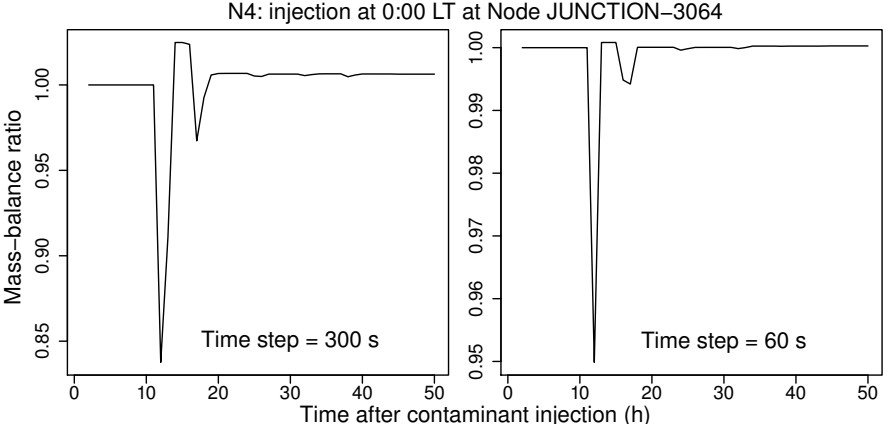

**Figure 5.** Examples of how the mass-balance ratio can vary during simulations. Results are for an injection at Node JUNCTION-3064 in Network N4. The location of Node JUNCTION-3064 is shown in Fig. A4.

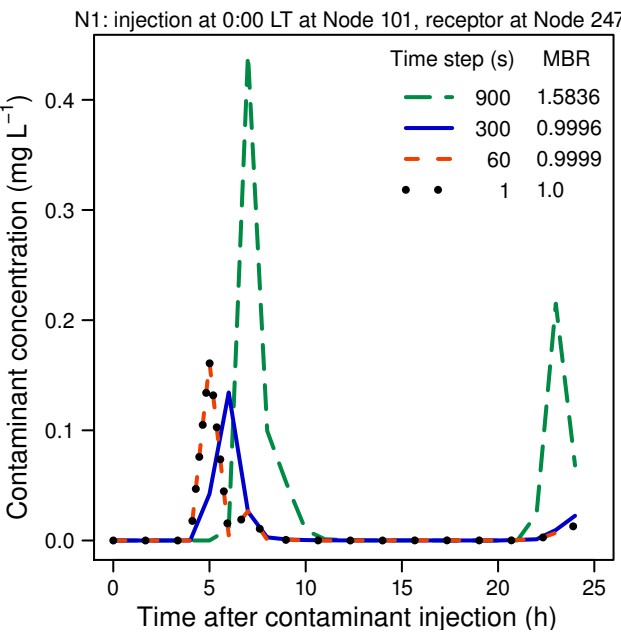

**Figure 6.** Influence of the water-quality time step on estimated contaminant concentrations at Node 247 in Network N1 following an injection at Node 101 at 0:00 LT. Locations of the nodes are shown in Fig. A1.



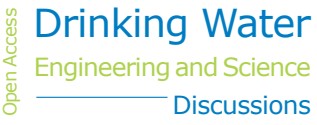

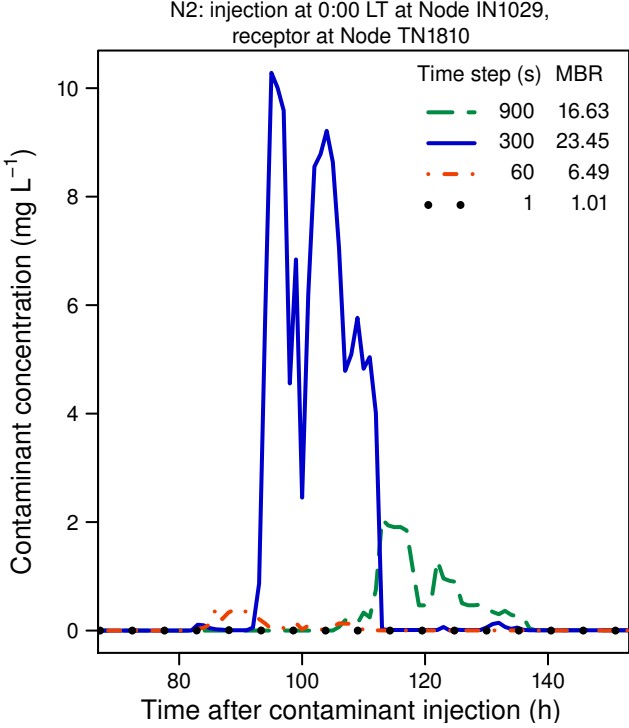

**Figure 7.** Influence of the water-quality time step on estimated contaminant concentrations at Node TN1810 in Network N2 following an injection at Node IN1029 at 0:00 LT. Locations of the nodes are shown in Fig. A3.

Using the time-driven method, contaminant concentrations were obtained at Node 200 in Network N3 after an injection at Node 100; they are compared in Fig. 8 for different water-quality time steps. Again, the timing and magnitude of the contaminant pulse change as the time step is reduced to 1 s, as is the case in the examples presented for Networks N1 and N2. The MBRs given in the figure are values at the end of a 168 h simulation.

Figure 9 compares contaminant concentrations obtained for a receptor node, TANK-12525, in Network N4 following an injection at Node JUNCTION-2514, using different water-quality time steps. Again, the simulations used the time-driven method. In this case, the timing of changes in concentration is similar for all the time steps used and there is a high degree of correlation between the results for the different time steps. However, concentration increases consistently as the time step decreases, unlike the trends in the previous examples. Concentrations appear to have approximately stabilized by a time step of

60 s, with little increase in concentrations occurring when the time step is reduced to 1 s. The MBRs given in the figure were determined at the end of a 168 h simulation.

    Statistics on the extent of mass imbalance at the end of simulations that used the time-driven method are provided in Table 2 for the four networks considered. The results in the table are based on independent injections at most nodes for each of the networks. For example, the statistics for Network N4 are based on independent simulations of injections done for most of

the 12,523 nodes in the network; a small fraction of the nodes were excluded, as discussed in the next paragraph. The table



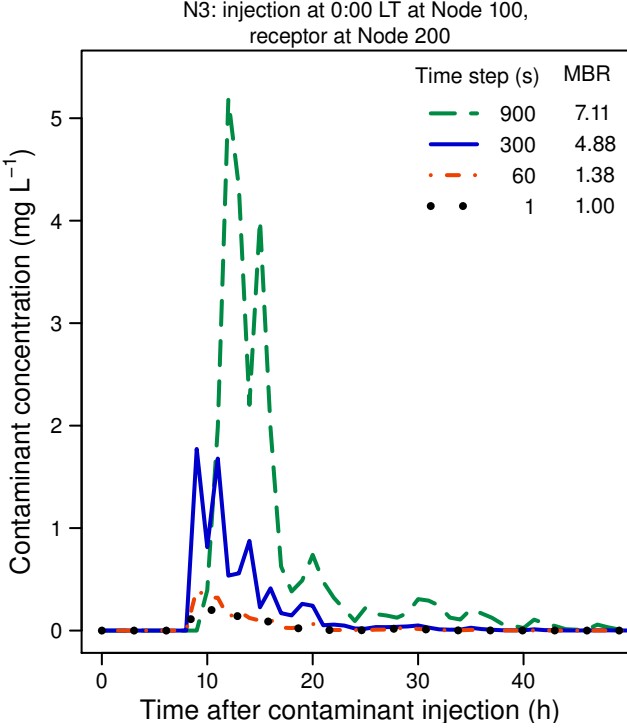

**Figure 8.** Influence of the water-quality time step on estimated contaminant concentrations at Node 200 in Network N3 following an injection at Node 100 at 0:00 LT.

provides the range in MBRs determined for each network for each of four water-quality time steps and the fraction of injection nodes for which there were imbalances above some thresholds (e.g., 1, 5, 10%).

MBRs equal to zero were obtained for injections at some nodes: the numerator in the MBR was zero because no mass was present in the pipes or tanks at the end of the simulation and no mass was removed during the simulation. Such cases can occur

when there is no flow at the time of injection (e.g., zero-demand nodes). When there is no flow at the time of injection, no contaminant mass is added to the water in the WDS and the injected mass is effectively lost, although the accounting process considers it to be mass injected when determining an MBR. Injection nodes for which an MBR was equal to zero at the end of a simulation were not included when determining the statistics shown in Table 2. For Network N4, two additional nodes (JUNCTION-9097 and -12348) also were excluded. These two nodes are in dead-end areas with no demands. Therefore, there

should be no flows in these areas. However, the network model had a small initial flow at these nodes, inconsistent with a lack of demands in the dead-end areas. For Network N3, five nodes had MBRs near 0.2 for all water-quality time steps; however, when the hydraulic and reporting time steps were changed to 1 min, the MBRs for the 1 min and 1 s water-quality time steps were near 1. MBRs near 1 were used for the five nodes when determining the statistics for Network N3 shown in Table 2.

Table 2 shows that as the water-quality time step decreases, the maximum MBRs for each network decrease towards 1.0

and the minimum MBRs generally increase. However, for Network N2 the minimum MBR increased only to 0.08 for the 1 s



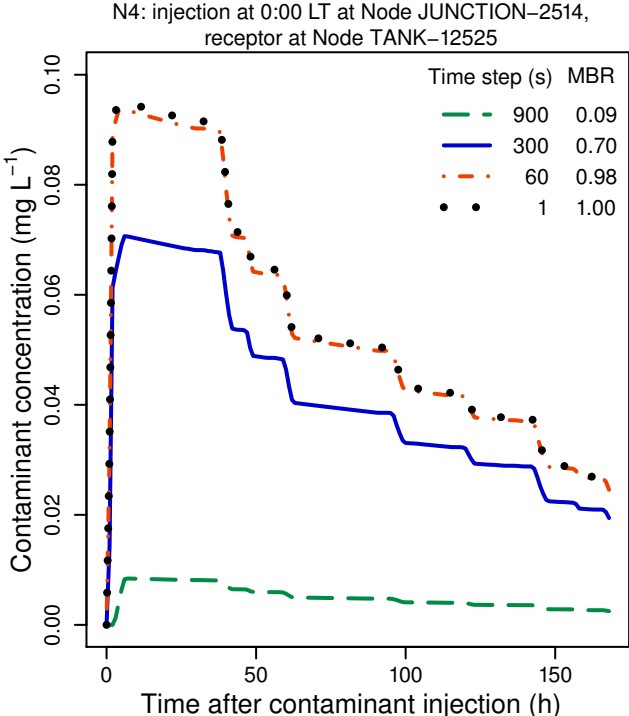

**Figure 9.** Influence of the water-quality time step on estimated contaminant concentrations at Node TANK-12525 in Network N4 following an injection at Node JUNCTION-2514 at 0:00 LT. Locations of the nodes are shown in Figs. A4 and A5.

time step and for Network N4 it reached only 0.83. For all four networks considered, the fraction of injection nodes with imbalances above the thresholds listed in the table decreases consistently as the time step decreases. For a time step of 1 s, only about 2, 1, and <1% of the nodes had imbalances greater than 1% for Networks N2, N3, and N4, respectively. There were no mass imbalances greater than 0.01% for this time step for Network N1 (excluding three nodes for which the MBR was zero).

5   This is in contrast to the sizable fraction of nodes in all the networks that have imbalances for a time step of 300 s, although the imbalances for Networks N1 and N4 are relatively minor for this time step, with only about 1 and 2% of nodes in these networks, respectively, having an imbalance greater than 10%.

For a small fraction of the injection nodes in Networks N2, N3, and N4, about 0.8, 0.5, and 0.2% of all nodes, respectively, the MBR did not change as the water-quality time step decreased or did not change so that the MBR converged toward 1.0.

10   The lack of convergence of the MBR for these nodes had limited influence on the statistics in Table 2; it did influence the values shown for minimum MBR for Network N2, particularly for a time step of 1 s. The non-convergence can be the result of several factors, including cases involving nodes in dead-end areas, as noted above. In addition, some cases had flows at the time of injection that were unexpected, given a lack of demands; the flows disappeared for subsequent time steps. Some problems appear to be related to the hydraulic solution; these were eliminated if a short (60 s) hydraulic time step was used.





**Table 2.** Statistics for mass imbalances.

| Network | WQTS (s) | MBR | | Injection nodes (%) with imbalances | | | |
|---|---|---|---|---|---|---|---|
| | | Max. | Min. | >1% | >5% | >10% | >50% |
| N1 | 1 | 1.00 | 1.00 | 0 | 0 | 0 | 0 |
| | 60 | 1.02 | >0.99 | 2 | 0 | 0 | 0 |
| | 300 | 1.15 | 0.97 | 38 | 6 | 1 | 0 |
| | 900 | 1.77 | 0.83 | 65 | 46 | 31 | 5 |
| N2 | 1 | 1.05 | 0.08 | 2 | 1 | 1 | 1 |
| | 60 | 6.49 | 0.06 | 39 | 29 | 18 | 5 |
| | 300 | 23.45 | <0.01 | 47 | 39 | 32 | 17 |
| | 900 | 31.16 | <0.01 | 52 | 49 | 44 | 24 |
| N3 | 1 | 1.03 | 0.99 | 1 | 0 | 0 | 0 |
| | 60 | 1.39 | 0.92 | 7 | 1 | -[a] | 0 |
| | 300 | 4.88 | 0.22 | 35 | 15 | 8 | 1 |
| | 900 | 15.74 | 0.08 | 57 | 36 | 25 | 6 |
| N4 | 1 | 1.01 | 0.83 | -[b] | -[c] | -[d] | 0 |
| | 60 | 1.04 | 0.67 | 1 | -[e] | -[c] | 0 |
| | 300 | 1.21 | 0.61 | 21 | 4 | 2 | 0 |
| | 900 | 2.60 | 0.09 | 44 | 28 | 19 | 1 |

*Note:* WQTS: water quality time step; MBR: mass balance ratio. Results do not include cases with
MBR = 0, which occurs for nodes with no flow at the time of injection. In addition, for Network N4, two
nodes also are excluded that are in dead-end areas with no demands, but that have small flows at the
beginning of the simulation. A total of three nodes are excluded for Network N1 (3%), 92 or 93 for Network
N2 (6%), depending on the time step, 55 or 56 for Network N3 (1%), again depending on the time step, and
404 for network N4 (3%). See text for discussion.
[a]Thirteen nodes. [b]Eleven nodes. [c]Two nodes. [d]One node. [e]Three nodes.

Mass balances are sensitive to the injection time used. An acceptable mass balance for a particular application for a given injection node does not guarantee an acceptable mass balance if hydraulic conditions are changed. As an example of the extreme changes that can occur when hydraulic conditions change, consider an injection at Node VN1263 in Network N2 (See location in Fig. A3), a 168 h simulation, and a 900 s water-quality time step. For a 1 h injection starting at 6:00 hours, the MBR at the end of the simulation was 467. For a 1 h injection starting at 0:00 hours, the MBR at the end of the simulation was 0.007. In the first case, much more mass was removed from the network by nodal demands during the simulation than was injected;





in the second case, very little mass remained in the system at the end of the simulation or was removed from the system by nodal demands during the simulation. In both cases the extreme values for the MBR are the result of errors in the estimated concentrations downstream from the injection location, too high in the first case, too low in the second. These errors resulted in the erroneous numerical generation or loss of constituent mass in the system.

The cases considered to this point used relatively short, 1 h injections. However, major mass imbalances also occur for injections with long durations. For example, with a 300 s water-quality time step, the largest MBR for any injection node in Network N2 for a 1 h injection at 0:00 hours is about 23; for 6 and 12 h injections at that time the largest MBRs are 1495 and 971, respectively. About 17, 22, and 22% of injection nodes have mass imbalances greater than 50% for the 1, 6 and 12 h injections, respectively. About 32, 39, and 35% have imbalances greater than 10%. The statistics for mass imbalance

are relatively insensitive to injection duration for this network. However, mass balances for a particular injection node can be sensitive to the injection duration. For example, for Node VN1263 in Network N2 the MBRs are 3.3, 1495, and 971 for injections at 0:00 hours with durations of 1, 6, and 12 h, respectively.

EPANET 3, the development version of EPANET (OpenWaterAnalytics, 2017b), also yields results with mass imbalances. Consider a case involving chlorine decay with both bulk and wall reactions and EPANET Example Network 1 (U.S. EPA,

2017a) with default input parameters, except for the water-quality time step. The network is very simple, with only nine junctions and 12 pipes. A chlorine mass imbalance of 0.85% (< 1%) was obtained for a water-quality time step of 900 s and a 24 h simulation. For time steps of 300, 60, and 1 s, the imbalances were 0.77, 0.58, and 0.49%, respectively. Mass imbalance was determined in EPANET 3 in the same manner as discussed in this paper except that the initial mass of chlorine in the system also was considered, as was the mass of chlorine lost due to chemical reactions. Execution time increased from about

0.02 s to about 0.03, 0.04, and 1 s when the time step was decreased from 900 s to 300, 60, and 1 s, respectively, an overall increase in execution time of about 50 fold. To obtain mass imbalances well below 0.1%, a time step well below 1 s may be needed, along with additional increases in execution time. This is an extremely small, simple network and large imbalances are not expected. For this example, the EPANET 3 code (OpenWaterAnalytics, 2017b) was compiled using the GNU Compiler Collection (GCC, 2017).

**4    Simulations with the preliminary event-driven approach**

The preliminary event-driven approach is discussed and results obtained using this approach are compared to those obtained using the time-driven approach.

**4.1    Background**

Changes in a WDS do not occur at regular time intervals. For example, the time required for a water parcel to move from node

to node in the system varies from pipe to pipe and also within a pipe as conditions change. In addition, some pipes can be short, have a high flow rate, and require only a short time for a water parcel to move through them. This transit time can be



too small to be practical for use as a water-quality time step in a simulation. A situation in which events (changes) occur at irregular intervals suggests using an event-driven simulation.

A preliminary event-driven algorithm is outlined here and used to obtain results for comparison with those provided by EPANET using the current time-driven approach. The event-driven approach used is similar to the Lagrangian event-driven

simulation method discussed in Rossman and Boulos (1996). Their event-driven method updates the state of water quality in the system only when a change occurs, in contrast to the current time-driven method in EPANET, which updates water quality across the entire network at fixed time steps. Various event-driven approaches have been presented previously, for example by Boulos et al. (1994, 1995). The preliminary event-driven algorithm discussed here is included as an option in the current version of TEVA-SPOT (TEVA-SPOTInstaller-2.3.2-MSXb-20170110-DEV) and is being made available to EPANET developers to

obtain community support and assistance with improving and evaluating the algorithm.

The event-based, water-quality routing algorithm used here moves homogeneous volumes of water (water parcels with a uniform concentration of a water-quality constituent) through a network. Nodes are processed in an arbitrary order as long as all inflow paths to a node have water parcels with a known constituent concentration. Mixing or combining of water parcels occurs at nodes based on the inflow rates of the links flowing into the nodes. Water parcels are combined if the absolute

difference between their concentrations is less than some specified amount (the quality tolerance), consistent with the approach used in EPANET 2. After parcels are combined at a node, any nodal demand is removed; the remaining water parcels then are split based on the flow rates of the links flowing from the nodes. These parcels are added to lists of parcels for the downstream links. Any volume in excess of the volume of a link is removed from the leading parcels and placed at the downstream node for further processing. Due to recirculating flows, situations can occur in which there are nodes for which constituent

concentrations have not yet been determined for all inflow links. In these cases, an incomplete parcel is created that has the volume that will be moved, but an unspecified concentration. These incomplete parcels are moved, combined, and split in the same manner as parcels for which constituent concentration has been determined; however, internal references are maintained that allow concentrations to be updated when parcels for which concentrations have been determined arrive at a node for which incomplete parcels were created. Flow reversals between hydraulic time steps are accommodated in the same manner as in

EPANET 2. The event-driven simulation method provides results that do not depend on the water-quality time step if it is equal to or shorter than the hydraulic time step. The method actually does not require an independent water-quality time step: the simulation is event driven as long as the hydraulic conditions do not change. Because by construction the method accounts for every individual water parcel, its resulting MBR will always be 1.0. An example illustrating the operation of the algorithm using a case with recirculating flow is provided in Appendix C.

**4.2    Discussion**

Concentrations obtained using EPANET's time-driven algorithm tend to converge to those obtained using the event-driven algorithm as the water-quality time step used in the time-driven algorithm decreases. For short water-quality time steps (e.g., 1 s) with the time-driven approach, the results for the two methods can be very similar and differences can be difficult to see in the plots used in this paper. Therefore, to better examine this convergence, least-squares fits were determined relating (1) the



**Table 3.** Least-squares fits of concentration results.

| Network | N | Case | $a$ | $b$ | adj-$R^2$ | Residual SD |
|---|---|---|---|---|---|---|
| N1 | 24 | TD300 | 0.2547 | 0.0074 | 0.0478 | 0.0273 |
|  |  | TD60 | 0.9985 | -1E-6 | 0.9999 | 0.0003 |
|  |  | ED3600 | 0.9995 | 0.00001 | 1 | 0.0001 |
| N2 | 60 | TD300 | -24.68 | 2.14 | -0.0169 | 3.385 |
|  |  | TD60 | 52.42 | 0.0334 | 0.3196 | 0.0911 |
|  |  | ED3600 | 0.944 | 0.0003 | 0.7657 | 0.0006 |
| N3 | 39 | TD300 | 6.159 | -0.0115 | 0.8866 | 0.1398 |
|  |  | TD60 | 1.515 | -0.00351 | 0.9866 | 0.0112 |
|  |  | ED3600 | 0.9997 | 0.0001 | 0.9999 | 0.0005 |
| N4 | 168 | TD300 | 0.7250 | 0.0016 | 0.9956 | 0.0011 |
|  |  | TD60 | 0.9840 | 0.0001 | 1 | 0.0001 |
|  |  | ED3600 | 0.9791 | 0.0018 | 0.9762 | 0.0035 |

*Note:* This table gives the parameters of a least-squares fit of the concentrations for the cases shown to the concentrations obtained using the time-driven algorithm with a water-quality time step = 1 s. Quantities $a$ and $b$ are the slope and intercept, respectively, of the least-squares line. Cases: TD300 = time-driven algorithm with a 300 s time step; TD60 = time-driven algorithm with a 60 s time step; ED3600 = event-driven algorithm with a 3600 s time step. N: number of hourly concentration values used. SD: standard deviation.

concentrations obtained using the time-driven approach with a water-quality time step of 1 s (TD1) and the concentrations obtained using the same approach with a 60 s time step (TD60), (2) TD1 and concentrations obtained using the time-driven approach with a 300 s time step (TD300), and (3) TD1 and the concentrations obtained using the event-driven approach with a 3600 s time step (ED3600). These least-squares lines have the form $\hat{y} = ax + b$, where $a$ and $b$ are the slope and intercept of the fitted least-squares line, $x$ is the value of TD1, and $\hat{y}$ is the fitted value of TD60, TD300, or ED3600, depending on which is being used. The results of fitting least-squares lines are shown in Table 3 for the four cases examined in Figs. 6 to 9.

The number (N) of hourly concentration values used to obtain the results shown in the Table 3 corresponds approximately to the number of hourly concentration values shown in the figures for the different networks. For Network N1, N was 24, covering the entire length of the simulation. For Network N2, it was 60, the length of the middle portion of the plot in Fig. 7. For Network N3, N was 39, the length of the period from Hour 1 in the simulation to Hour 40 (see Fig. 8). For Network N4, results for the entire 168 h simulation were used. The water-quality tolerance in the simulations used to obtain the concentrations needed for the analysis presented in the table was 0.01, except for the event-driven simulations for Network N4, for which 0.1 was used.

If the concentrations obtained using the time-driven method with a 1 s water-quality time step are identical to those obtained for one of the other cases, the slope of the least-squares line relating the concentrations will be 1, the intercept will be 0, the adjusted $R^2$ will be 1, and the residuals for the fit will all be 0. From Table 3, the results for Networks N1, N2, and N3 show that

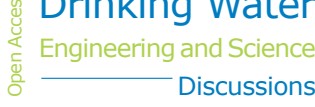



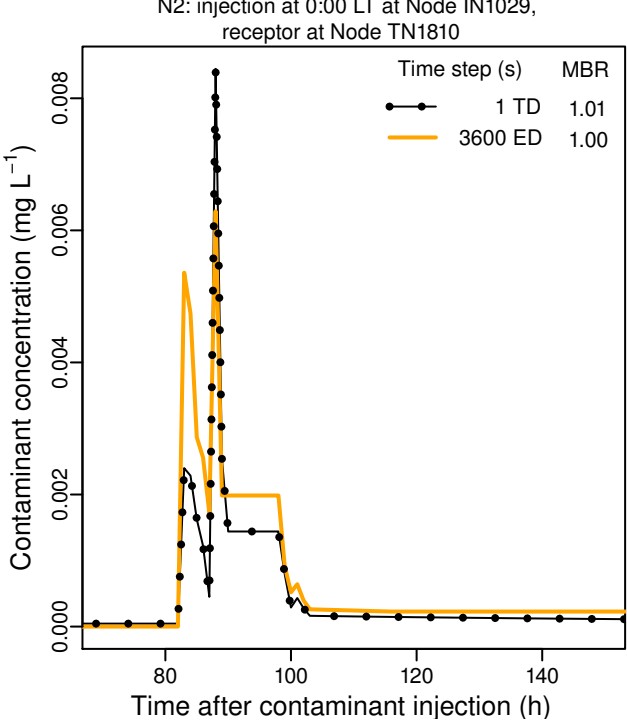

**Figure 10.** Estimated contaminant concentrations at Node TN1810 in Network N2 following an injection at Node IN1029 obtained with the time-driven (TD) and event-driven (ED) water-quality algorithms. Locations of the nodes are shown in Fig. A3.

of the three cases considered, the concentrations for TD1 are closest to those for ED3600. Also, the agreement improves going from TD300 to TD60 to ED3600. This improvement is what would be expected if the results for the time-driven approach are converging to those for the event-driven approach as the water-quality time step used for the time-driven approach decreases. For Network N4 there is a high degree of correlation between the concentrations obtained using the time-driven method with different time steps (cf. Fig. 9) and the quality of the fit is similar for all three cases considered. However, because the magnitude of the concentrations increases as the time step decreases, the slope of the least-squares line increases going from TD300 to TD60.

The results in Table 3 for Network N2 indicate a less-than-perfect correlation between the concentrations obtained with the time-driven algorithm used in EPANET and a water-quality time step of 1 s and those obtained with the event-driven algorithm and a time step of 3600 s. The concentrations obtained with the two approaches are compared in Fig.10 (which uses an expanded vertical scale compared to the one used in Fig. 7). The results obtained for the two simulations are noticeably different. However, compared to the differences between the results obtained with the time-driven approach using a 1 s time step and those obtained using longer time steps, the differences are quite minor, as can be seen from a comparison of Figs. 7 and 10.





Overall, the results presented here demonstrate that substantial mass imbalances can occur during EPANET water-quality simulations. Such mass imbalances tend to disappear and significant changes in constituent concentrations can occur as the water-quality time step becomes small. Also, these constituent concentrations tend to converge to those obtained with the event-driven simulation method, which conserves constituent mass.

5    The preliminary event-driven algorithm discussed here currently addresses only those constituents that behave as tracers. The algorithm needs to be expanded to consider constituent decay. The algorithm also needs to be evaluated using a wider range of networks and cases. The accuracy, storage requirements, and computation time for other types of water-quality modeling problems, such as source tracing, water age, and chlorine decay need to be examined. Preliminary results are presented here to help motivate additional efforts to improve water-quality simulations in EPANET.

10    The results presented here indicate that, in general, a water-quality time step of 1 s may be necessary to obtain acceptable mass-balance results when using the time-driven approach in EPANET. For large networks, such a time step can require considerable computational effort. Statistics for execution times for TEVA-SPOT, using EPANET and the time-driven algorithm, are provided in Davis et al. (2016) for several network models, including Network N4 (BWSN, called Network E3 in the reference) for a 1 s water-quality time step. Results in the reference are for a subset of the nodes considered here, only those with a non-zero demand, and include some additional computations beyond those used for this paper, but demonstrate substantial execution times. For a single injection, execution times were about 70 min using a 2.3 GHz processor. For injections at all non-zero demand nodes for the network, the execution time was about 16 days for a server with four such processors using 32 cores, with 32 simultaneous simulations being performed. The event-driven algorithm is not fully developed; however, because a water-quality time step as long as the hydraulic time step can be used, it is expected to generally require less computational effort than the time-driven algorithm if conservation of mass is required. Table 4 compares execution times for the time-driven and event-driven algorithms for examples used in this paper, with a time step of 1 s for the time-driven algorithm and 3600 s for the event-driven one. The execution times were obtained using a single 2.8 GHz processor. In general, for the cases considered here, the event-driven approach requires substantially shorter execution times than the time-driven approach in EPANET when a 1-s water-quality time step is used.

## 5    Conclusions

As the examples presented here illustrate, the current version of EPANET can produce results for which the mass of a water-quality constituent is not conserved. Significant mass imbalances can occur when modeling water quality, even for water-quality time steps considerably shorter than those commonly used with EPANET. These mass balances are associated with inaccurate estimated constituent concentrations.

30    Substantial mass imbalances can occur at the beginning of a simulation, but be reduced or eliminated as the simulation proceeds. Therefore, if unacceptable mass imbalances occur for a short simulation, a longer simulation time may be needed.

Although mass imbalances can be reduced or eliminated by decreasing the size of the water-quality time step, sufficient reductions may not be practical. As noted above, the default water-quality time step in EPANET is 300 s and longer time steps



**Table 4.** Comparison of execution times for the time-driven and event-driven algorithms.

| Network | Injection location | Execution time (s) | |
|---|---|---|---|
| | | TD[a] | ED[b] |
| N1 | All 97 nodes | 259 | 24 |
| N2 | Node IN1029 | 970 | 294 |
| | Node VN1263[c] | 326 | 670 |
| N3 | Node 100 | 2990 | 220 |
| | Node 300 | 1440 | 17 |
| N4 | Node JUNCTION-2514 | 1780 | 15 |
| | Node JUNCTION-3064 | 1760 | 17 |

*Note:* All simulations were done using a 2.8 GHz processor. The simulation
durations were 168 h, except for Network N1, for which they were 24 h. All
times are rounded to three significant figures or to the nearest whole second.
TD: time driven; ED: event driven.
[a]Water-quality time step = 1 s.
[b]Water-quality time step = 3600 s.
[c]Quality tolerance = 1.0 mg L$^{-1}$.

are used in some applications. However, in some cases, as shown here, use of a time step as short as 60 s can result in significant errors; a time step less than 60 s may be necessary to obtain acceptable results and in some cases a time step of 1 s does not eliminate mass imbalances.

Results from the current version of EPANET tend to converge to those obtained with our preliminary event-driven water-quality algorithm as the water-quality time step used with the current version of EPANET is reduced. The event-based algorithm for water-quality routing provides results that conserve constituent mass and that are independent of the water-quality time step if it is less than or equal to the hydraulic time step. Given this independence of the size of the water-quality time step, the new algorithm may not only be more accurate, but also more economical to use than the one currently included in EPANET if mass conservation is required.

The event-driven algorithm used here is under development. It is available as an option for water-quality modeling in TEVA-SPOT. Additional refinement of the approach is needed; in particular, it needs to be modified to consider non-conservative constituent behavior.

EPANET water-quality simulations are widely used by water utilities. The results presented here should be of value to utility managers and engineers; they allow users of such simulations to better understand an important potential limitation of these simulations. The results also allow them to understand how these simulations can be improved.



## 6   Recommendations

On the basis of results presented here, we recommend the following:

1. The default water-quality time step for EPANET with the current time-driven water-quality algorithm should be 60 s. The time step should not exceed 300 s.

2. A capability to produce reports on the mass balance of water-quality constituents needs to be added to EPANET.

3. When a capability to obtain an evaluation of mass balance is available, the water-quality time step should be selected so that acceptable mass balances are obtained.

4. The water-quality algorithm used in EPANET needs to be replaced with one that conserves mass and provides accurate concentration estimates.

10   *Code and data availability.*   Models for Networks N1, N2, and N4 are available at Data.gov and can be found by searching using this paper's title. The model for Network N3 is proprietary and cannot be shared. The preliminary, Lagrangian event-driven algorithm discussed in this paper is available for inspection and (hopefully) collaboration at https://github.com/ttaxon/EPANET/tree/flow-transport-model.

## Appendix A:  Network maps

Schematics of Networks N1, N2, and N4 are shown in Figs. A1, A3, and A4, respectively. Additional detail for Network N1 is
15   shown in Fig. A2 and for Network N4 in Fig. A5. Because it contains confidential information, the schematic for Network N3 cannot be provided. The nodes identified in the figures are used in examples discussed in this paper.



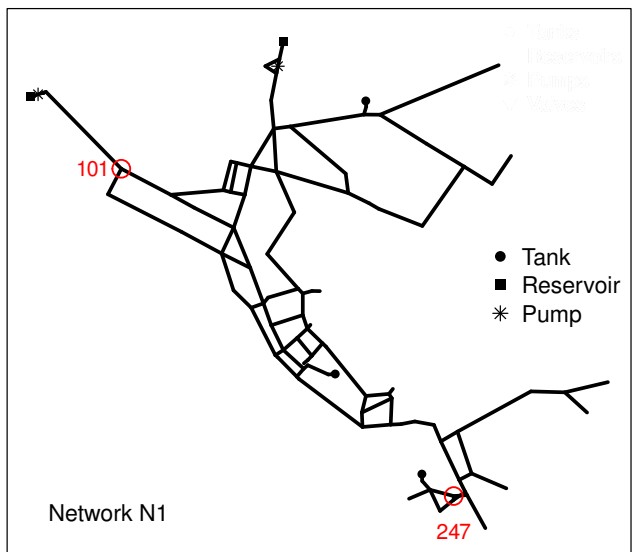

**Figure A1.** Network N1.

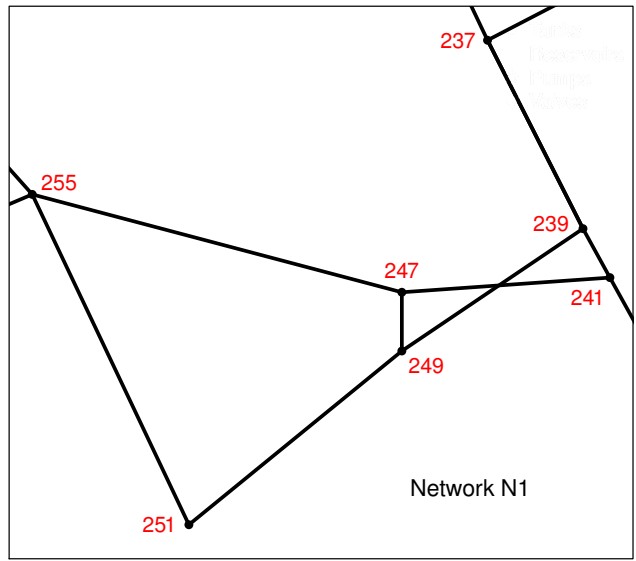

**Figure A2.** Detail in Network N1.



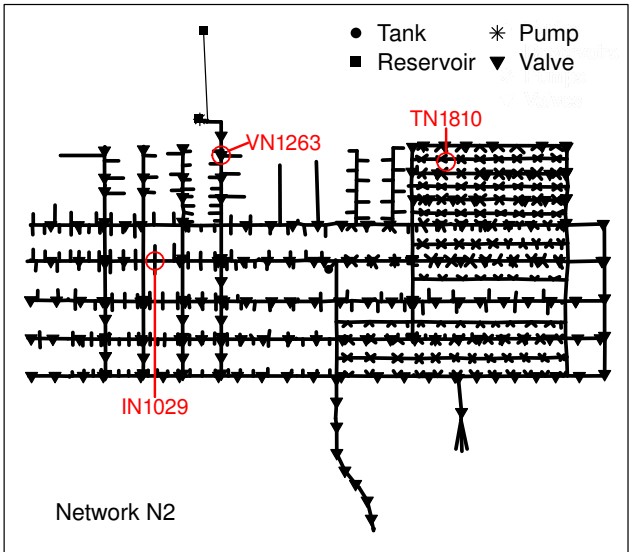

**Figure A3.** Network N2.

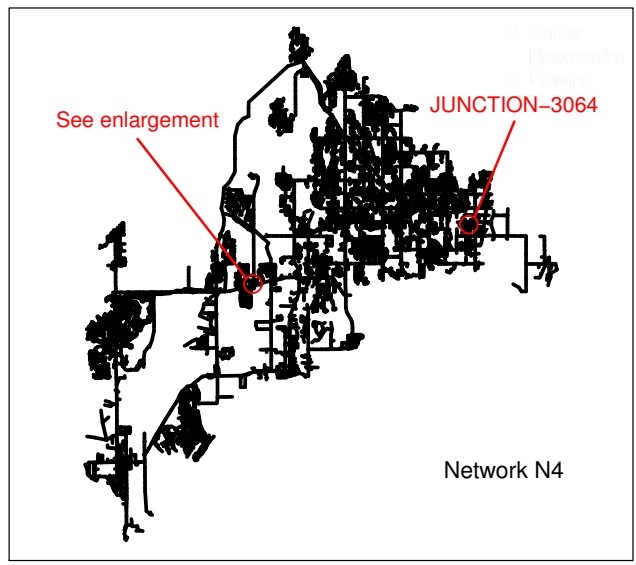

**Figure A4.** Network N4. See Fig. A5 for an enlargement showing detail in the highlighted area.



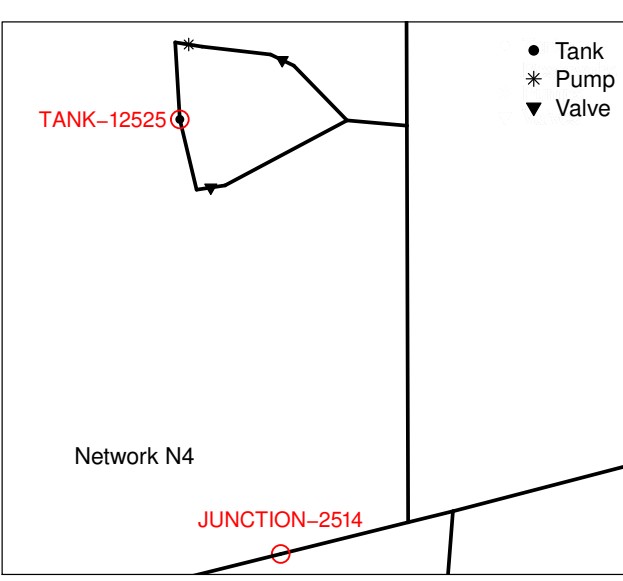

**Figure A5.** Detail in Network N4.



**Appendix B: Example using the time-driven approach**

This appendix presents an example illustrating the operation of the time-driven algorithm used in EPANET and, in particular, shows how the algorithm can fail to conserve constituent mass. The example uses the portion of Network N1 shown in Fig. A2. The example presented here is designed to demonstrate how mass imbalance can occur and is not a suggestion that

the parameters used in the example would actually be used.

**B1    Water-quality routing with the time-driven approach**

Water-quality routing in EPANET uses the function transport(), which in turn uses several other functions to accumulate constituent mass and water volume at nodes and to create new water parcels in outflow links. Accumulating, releasing, and updating are each done at the same time at all nodes in a network (a time-driven approach). The functions listed below are used

in the order shown:

**accumulate()** This function accumulates constituent mass (MassIn) and inflow volume (VolIn) at nodes and computes nodal constituent concentrations, which are stored for later use.

**updatenodes()** This function updates concentrations at all nodes. It does not consider additions from any sources of water-quality constituents at nodes.

**sourceinput()** If appropriate, this function accounts for any contributions of constituent mass from sources. (No sources are used in the example presented here.)

**release()** This function creates new parcels in outflow links for all nodes.

**updatesourcenodes()** This function updates water quality at source nodes using results from sourceinput(). (Not relevant for this example.)

The next section illustrates the application of the relevant functions using an example based on Network N1.

**B2    Example**

For the example presented in this appendix. a total of $0.5 \ \mathrm{kg}$ of a water-quality constituent was added at a constant rate at Node 249 of Network N1 during the first hour of a 24 h simulation. Water-quality and hydraulic time steps of 3600 s were used along with constant nodal demands. The quality tolerance was $0.01 \ \mathrm{mg \, L^{-1}}$.

The initial conditions in the portion of Network N1 (Fig. A2) used in this example are shown in Fig. B1. They correspond to those at hour two in the simulation. The example uses seven nodes (237, 239, 241, and so on). The italicized numbers below or to the right of the links connecting nodes are the link numbers (273, 275, etc.). There are demands at Nodes 239 and 247. The volumes of water moved during a water-quality time step are shown above or to the left of the links; the arrows indicate the direction of flow for the time step shown. For example, $5.941 \ \mathrm{m^3}$ of water are moved from Link 281 to Node 247 during

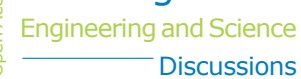

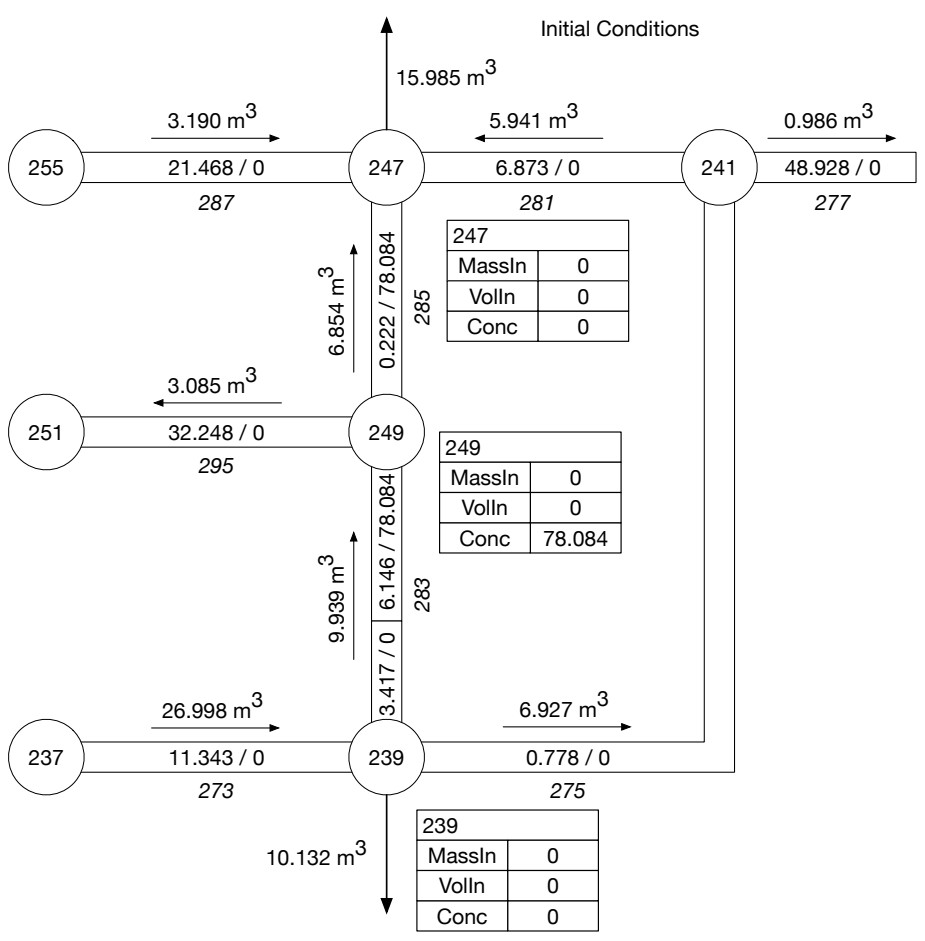

**Figure B1.** Example: Initial conditions.

the time step and 15.985 $m^3$ of water are removed by demands at Node 247 during the time step. The volumes of water parcels and the concentrations of a water-quality constituent present in parcels are shown. For example, Link 281 has one water parcel with a volume of 6.873 $m^3$ and a concentration of zero (shown as 6.873/0). Link 283 has two parcels; the leading parcel has a volume of 6.146 $m^3$ and a concentration of 78.084 $mg\,L^{-1}$ (6.146/78.084) and the trailing parcel has has a volume of 3.417 $m^3$

5   and a concentration of zero (3.417/0). The link volume is equal to the sum of volumes of the water parcels in the link; Link 283 has a volume of 6.146 + 3.417 = 9.563 $m^3$. The small tables in the figure provide initial values of MassIn, VolIn, and constituent concentrations for Nodes 239, 247, and 249. In these tables the units for MassIn, VolIn, and concentration are $mg$, $m^3$, and $mg\,L^{-1}$, respectively.

  Conditions after **accumulate()** has been called are shown in Fig. B2. The volumes and concentrations of parcels remaining

10   after nodal inflows have been removed from links are shown. For some links the volume removed exceeds the volume of the

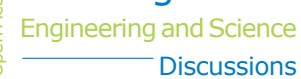

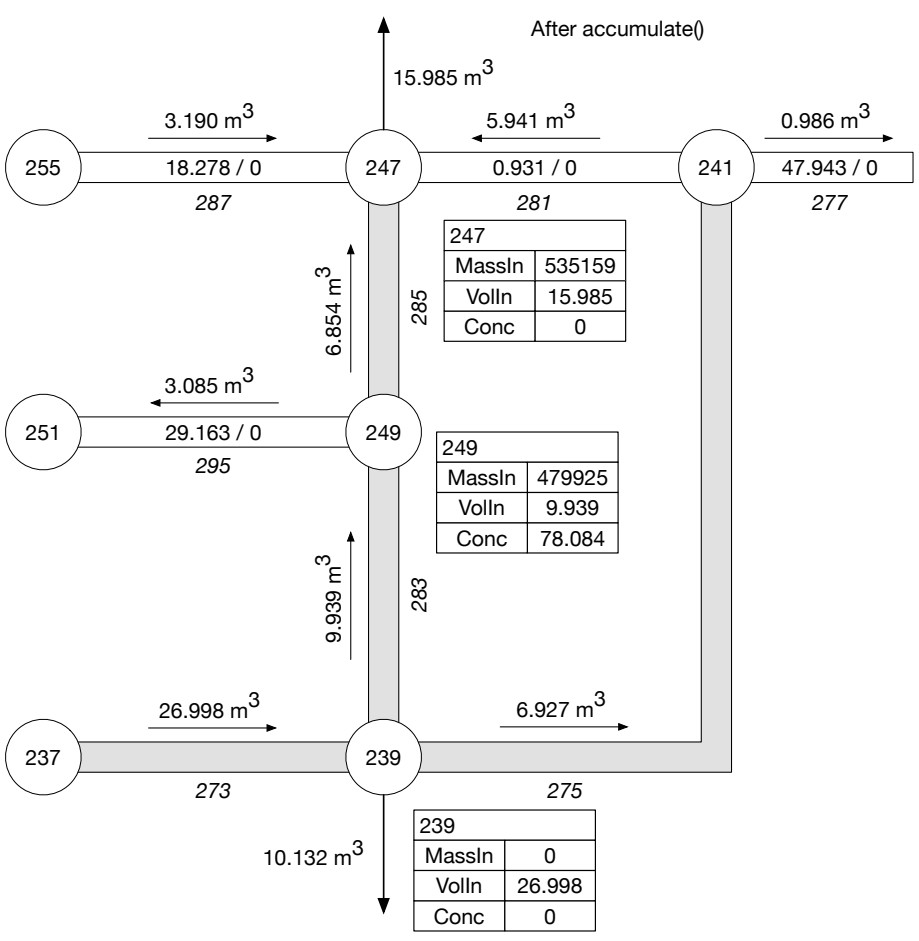

**Figure B2.** Example: Conditions after accumulate().

link and no parcels remain in the links. These links are shaded in the figure. When the volume being moved exceeds the volume of a link, the contribution from the link to VolIn for the downstream node is determined using the volume moved. For example, VolIn for Node 239 is 26.998 $m^3$, the volume being moved on Link 273, not the volume of the link, which is 11.343 $m^3$. Values of MassIn for downstream nodes are determined by the volume moved in a time step, not the volume of the water parcels in the links providing inflows. When the volume moved is greater than the link volume, the extra volume that is moved to the downstream node has the same concentration as the trailing water parcel in the link. For example, for Node 247 the constituent mass coming from Link 285 is determined using the volume moved (6.854 $m^3$) and the concentration (78.084 mg L$^{-1}$) of the water parcel in the link (the only parcel in the link and, therefore, also the trailing parcel), which gives a contribution to MassIn for the node of about 0.535 kg. The concentrations of the water parcels in Links 281 and 287 are zero and the links do not contribute any mass to MassIn for Node 247. The value of MassIn in the table for Node 247 (535,159 mg) is slightly different



from the product of VolIn and the concentration due to rounding. Note that the value for MassIn for Node 247 (0.535 kg) is larger than the constituent mass added to the network (0.5 kg). Excess mass has been generated at the node; a mass gain of about 0.518 kg has occurred, given that about 0.535 kg has been added to MassIn for Node 247 and not 0.017 kg, the mass of constituent in the parcel in Link 285, which has a volume of 0.222 m$^3$ and a concentration of 78.084 mg L$^{-1}$. Note that the value of VolIn for Node 247 is the sum of the water volumes moved from the three inflow links (281, 285, and 287).

For Link 283 the link volume is also less than the volume of water being moved in the time step (9.563 vs 9.939 m$^3$). The leading water parcel flows into Node 249, contributing a constituent mass of about 0.480 kg (6.146 m$^3$ multiplied by 78.084 mg L$^{-1}$). To provide the necessary volume being moved to the node, an extra 0.376 m$^3$ is added to the trailing parcel and given a concentration of zero, the same as concentration of the trailing parcel. The second parcel adds no constituent mass to Node 249; the value for MassIn for the node is about 0.480 kg, coming entirely from the leading water parcel on the link.

No constituent mass is accumulated at Node 239. Although the volume of water being moved in Link 273 is larger than the volume of the link, the concentration of the added volume is zero.

When the accumulate step is completed, values for MassIn and VolIn have been computed for all nodes in the network. Values for these quantities are shown in Fig. B2 for Nodes 239, 247, and 249. MassIn is zero for all other nodes. The initial conditions show that the water-quality constituent was present only in Links 283 and 285.

In the next step in the water-quality routing process, **updatenodes()** is called to update nodal concentrations. The updated concentrations for Nodes 239, 247, and 249 are shown in the tables in Fig. B3. The concentrations are determined using MassIn and VolIn for each node. For example, the concentration for Node 249 is 48.289 mg L$^{-1}$, obtained by dividing 479,925 mg by 1000 times 9.939 m$^3$.

In the final step in this example, a call to **release()** creates new water parcels on the outflow links for all nodes. Conditions after these parcels are created are shown in Fig. B4. New parcels are combined with parcels already present if the difference between their concentrations is less than the quality tolerance (0.01 mg L$^{-1}$). For example, a new parcel with a volume of 5.941 m$^3$ was added to Link 281 and combined with the 0.931 m$^3$ parcel already on the link because both parcels have a concentration of zero. New parcels with concentrations of 48.289 mg L$^{-1}$ are added to Links 285 and 295, using the concentration

for Node 249. The volume of the parcel for Link 285 is 0.222 m$^3$, the volume of the link. However, the volume released to the link is 6.854 m$^3$; the difference between this volume and the volume of the link (6.854 - 0.222 = 6.632 m$^3$) is effectively lost. Because the concentration associated with this volume is 48.289 mg L$^{-1}$, a mass loss of about 0.320 kg (48.289 times 6.632 divided by 1000) occurs. Adding the mass gain of 0.518 kg in the accumulate step, there is a net mass gain at this point in the simulation equal to about 0.2 kg. Given the constituent mass removed by demands from Node 247 (about 0.54 kg; 15.985 m$^3$

of water with a constituent concentration of 33.479 mg L$^{-1}$) and the mass in Links 285 and 295 (about 0.16 kg), the MBR for the network at this point is about 1.4 (0.54 plus 0.16 divided by 0.5). The values for MassIn and VolIn shown in the tables in Fig. B4 are retained from the updatenodes step; they have no meaning at this point and are reset to zero at the beginning of the next time step.

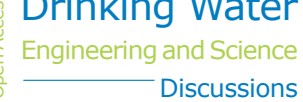



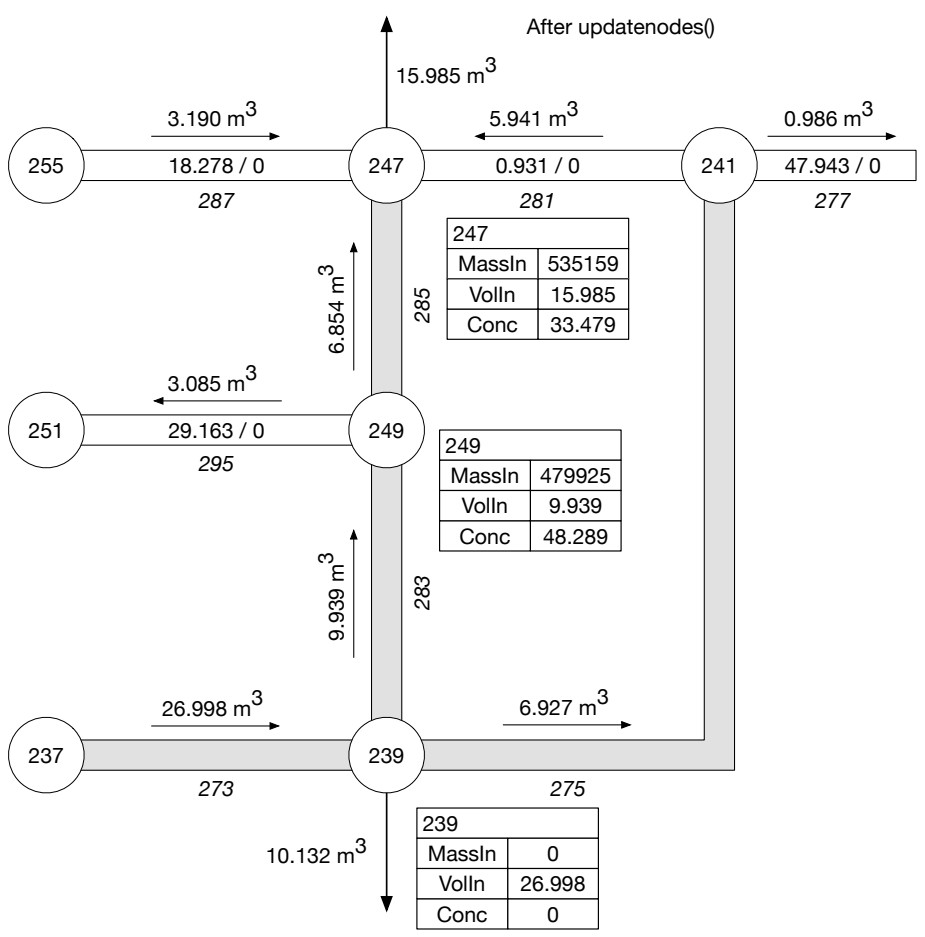

**Figure B3.** Example: Conditions after updatenodes().

## B3 Discussion

As the example presented here illustrates, the accumulate step can result in the generation of constituent mass and the release step can result in the loss of mass. The condition necessary (but not sufficient) for generating or losing mass is a link with a volume less than the volume of water being moved in a water-quality time step. If the concentration of a water-quality constituent present in the network does not vary spatially, the mass generated in the accumulate step will equal the mass lost in the release step and there will be no net change in mass. However, if there is a spatial gradient in concentration so that the concentration of the excess volume generated during the accumulate step is different from the concentration associated with the volume of water lost in the release step, a net change in mass can occur. The net change can be either positive or negative: net mass can be generated or lost.

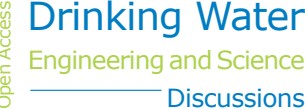

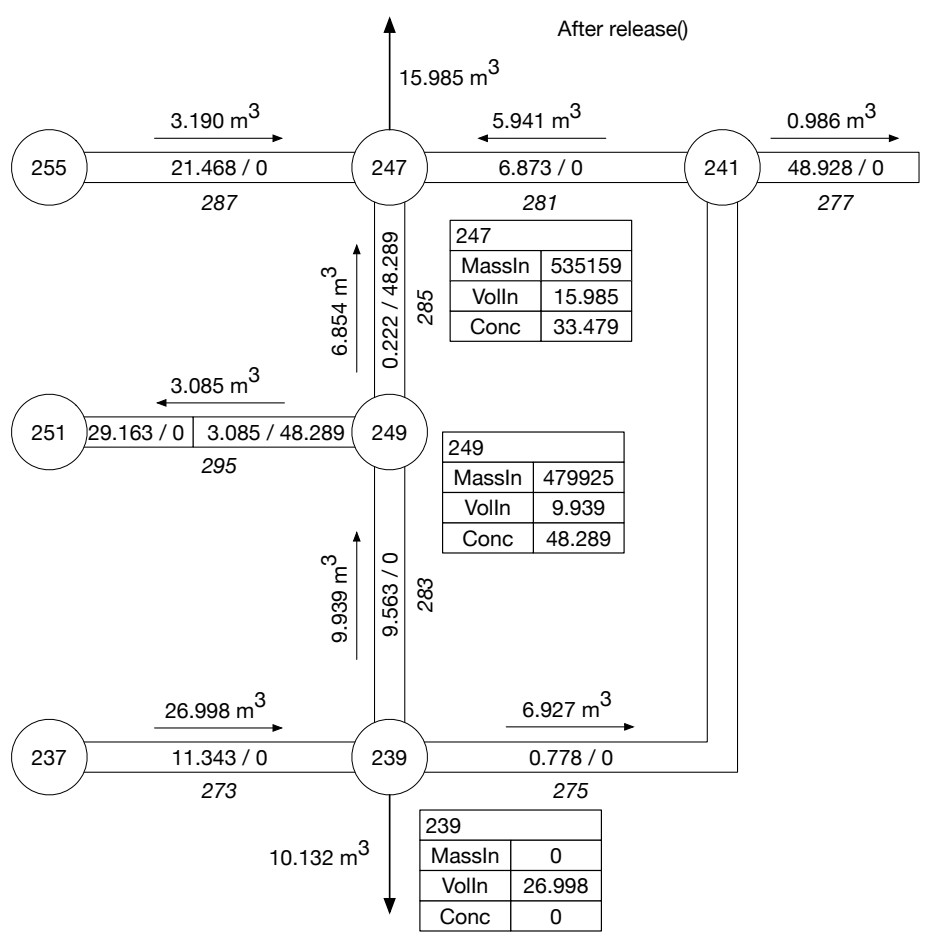

**Figure B4.** Example: Conditions after release().

Let $V_L$ be the volume of a link, $V_M$ be the volume of water moved in the link during a water-quality time step, and $V_M$ be greater than $V_L$. Let $C_A$ be the concentration of a constituent used for the extra volume for this link added in the accumulate step and $C_R$ be the concentration of the constituent used for the lost volume for this link in the release step. Then the net change in constituent mass ($\Delta M$) for the link for the time step is given by

$$5 \quad \Delta M = C_A(V_M - V_L) - C_R(V_M - V_L) = (C_A - C_R)(V_M - V_L) \tag{B1}$$



Note that this relationship applies only when $V_M > V_L$. The volume of water being moved in a time step is $QT$, where $Q$ is the flow rate on the link for the time step and $T$ is the length of the water-quality time step. Then,

$$\Delta M = (C_A - C_R)(QT - V_L) \tag{B2}$$

The volume of a link can be small; for a valve or pump it is zero. When the link volume is very small relative to $V_M$ the mass change for the link for the time step is proportional to the difference between the concentrations for the accumulate and release steps, the flow rate, and the water-quality time step. As the water-quality time step becomes small, the net mass change also become small.

For the example presented in this appendix, $C_A$ = 78.084 mg L$^{-1}$ and $C_R$ = 48.289 mg L$^{-1}$. The difference between the volume moved in Link 285 and the link volume is 6.854 m$^3$ - 0.222 m$^3$ = 6.632 m$^3$. Therefore, using Eq. (B1), the value for $\Delta M$ is (78.084 - 48.289)(6.632)/1000 = 0.198 kg. For a net mass gain of this amount the MBR is about 1.4, as noted above. The flow rate on Link 285 is small, 6.685 m$^3$ per hour or about 0.002 m$^3$ s$^{-1}$. If the flow rate were an order of magnitude larger, the net mass gain would about 2 kg and the MBR would be about 5. Substantial net mass changes are possible for a single link for a single water-quality time step. Note that when $C_R > C_A$, net mass losses will occur and the MBR will be less than one.

**Appendix C: Example using the event-driven approach**

This appendix presents a simple example to illustrate the application of the event-driven algorithm. The example also illustrates how the algorithm addresses cases with recirculating flows. The network used in the example is shown in Fig. C1. It has five nodes (A to E), with connecting links (labeled A-B, for example), Link D-E is a pump. Details of the network downstream (to the right) of Node D are not shown in the figure. There are demands at Nodes B, C, and E. The volumes shown above the links are the volumes of water moved during a water-quality time step; the arrows show the direction of flow. For example, 76 m$^3$ of water are moved from Link A-B to Node B and 20 m$^3$ of water are removed by demands at Node B during the time step. Each link can have one or more water parcels, which are volumes of water with uniform concentrations. The volumes of the parcels and the concentrations of a water-quality constituent present in the parcels are shown; for example, in Link E-B there are two parcels, one has a volume of 1700 m$^3$ and a concentration of zero (labeled as 1700/0) and the second has a volume of 100 m$^3$ and a concentration of 10 mg L$^{-1}$ (labeled as 100/10). The volume of a link equals the sum of the volumes of water parcels in the link. For example, Link E-B has a volume of 1800 m$^3$, equal to the sum of 100 m$^3$ plus 1700 m$^3$. Note that the volume of Link D-E, a pump, is zero. The quality tolerance is 0.01 mg L$^{-1}$; adjacent water parcels whose concentrations differ by less than this amount are combined.

As shown in Figure C1, there is an inflow to Node B from Link A-B with a concentration of zero. There is also an inflow from Link E-B; however, the concentration for that inflow has not yet been determined and the inflow cannot be processed. Consequently, the state of the inflow from Link E-B is labeled as 4/? in the small table in the figure, indicating that the concentration has not yet been determined. It may seem obvious that the concentration of the inflow from Link E-B should be

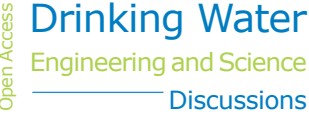

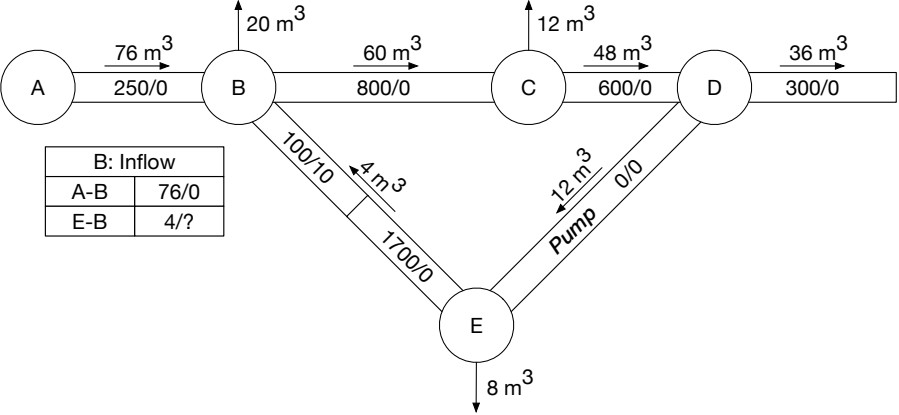

**Figure C1.** Example: Network at the beginning of the time step. (The 36 m$^3$ of water leaving Node D goes to a downstream portion of the network not shown in the figure.)

10 mg L$^{-1}$, given that this is the concentration of the leading parcel in the link. However, in general, the concentration of the inflow from Link E-B will be affected by the volume of the inflow. More than one water parcel might need to be combined to provide the needed inflow volume and the operations required to determine if any combining of parcels is necessary and to determine the resulting concentration have not yet been carried out. Therefore, to accommodate this situation, an incomplete

parcel representing the merging of inflows from Links A-B and E-B is created that has an unspecified concentration but a known volume. This incomplete parcel maintains references to the parcels that merged to form it. The parcel (called Seg1) is then moved through Node B with the result shown in Fig. C2, which illustrates how water parcels from Links A-B and E-B are combined at Node B to form Parcel Seg1, which is then split into Parcels Seg2 and Seg3 to satisfy the nodal demand and the outflow from the node. Node B's demand list has a reference to Parcel Seg2, which shows that of the parcel's total

volume of 20 m$^3$, the parcel from Link A-B contributed 19 m$^3$ and the parcel from Link E-B contributed 1 m$^3$; the ratio of the contributions to Parcel Seg2 (and Seg3) is 19:1, the same as the ratio of inflows from Links A-B and E-B. Parcel Seg4 is created and added as the trailing segment in Link B-C. After Parcel Seg4 is added to Link B-C, 60 m$^3$ of water are removed from the leading water parcel in the link; the leading parcel in the link then has a volume of 740 m$^3$ (and a concentration of zero, shown as 740/0 in Fig. C2), maintaining the total volume of the parcels in the link equal to the link's volume, namely

800 m$^3$.

At this point Parcel Seg1 is no longer referenced directly by any link or node. However, it has internal references so that when the concentration of the water parcel arriving at Node B from Link E-B has been determined, the parcel can be completed. When the parcel is completed, the result will cascade to its children, namely Seg2 and Seg3.

Adding Parcel Seg4 with a volume of 60 m$^3$ to Link B-C results in the same volume of water being moved to Node C from

the leading parcel on Link B-C. Twelve cubic meters of water are removed from that parcel and placed on Node C's demand list; the remaining 48 m$^3$ of water from the parcel are added to Link C-D. At Node D, 48 m$^3$ of water are removed from the



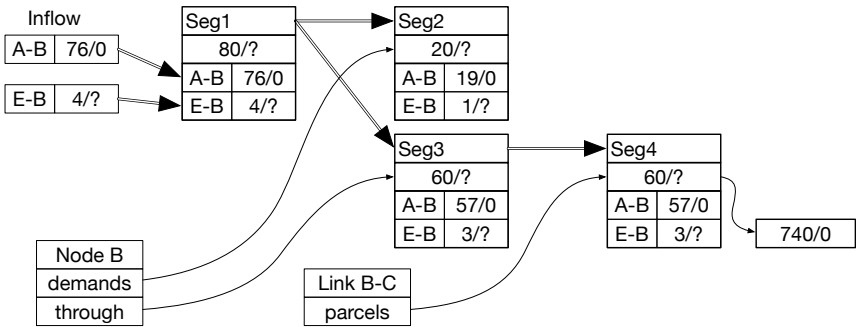

**Figure C2.** Example: Creation of incomplete water parcels.

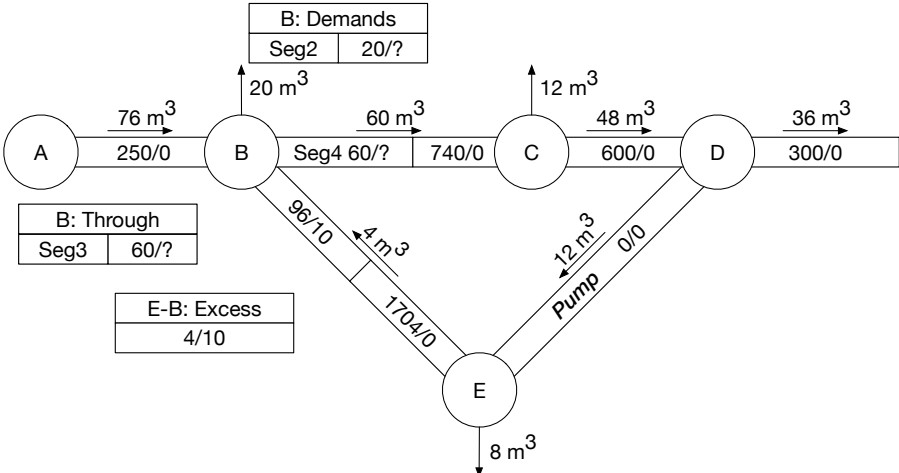

**Figure C3.** Example: Interim status of the network.

link, with the parcel that is removed being split: 36 m$^3$ leave Node D to be moved through the rest of the network (not shown) and 12 m$^3$ are placed on the trailing end of Link D-E. At Node E, 8 m$^3$ of water are removed and placed on Node E's demand list. A parcel with a volume of 4 m$^3$ is added at the trailing end of Link E-B, which causes 4 m$^3$ of the leading parcel in Link E-B (with a concentration of 10 mg L$^{-1}$) to be sent to Node B. Because the concentration of the 4 m$^3$ parcel added to Link E-B

5   is the same (zero) as the concentration of the trailing 1700 m$^3$ parcel, the two are combined to yield a parcel with a volume of 1704 m$^3$ and a concentration of zero. The situation at this point is shown in Fig. C3.

When the leading parcel from Link E-B arrives at Node B, incomplete Parcel Seg1 can be completed because its concentration can now be determined. The concentrations of all the other incomplete parcels shown in Fig. C2 also can be determined and updated values for these concentrations are shown in Fig. C4. The concentration of Parcel Seg1 (0.5 mg L$^{-1}$) is the concentra-

10  tion of the blended parcels coming from Link A-B (76 m$^3$ with a concentration of 0) and Link E-B (4 m$^3$ with a concentration

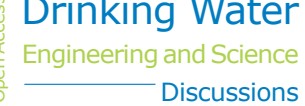



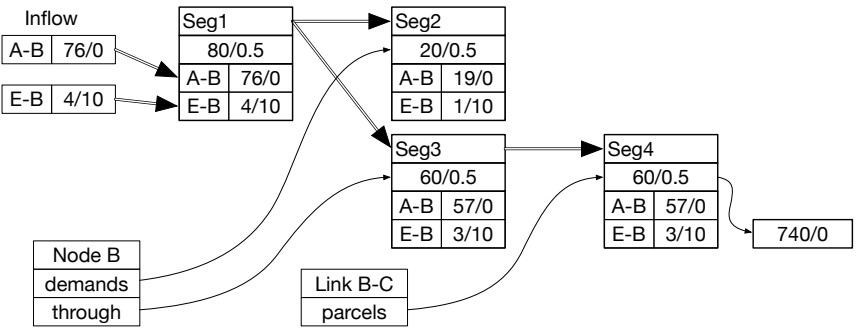

**Figure C4.** Example: Completion of incomplete water parcels.

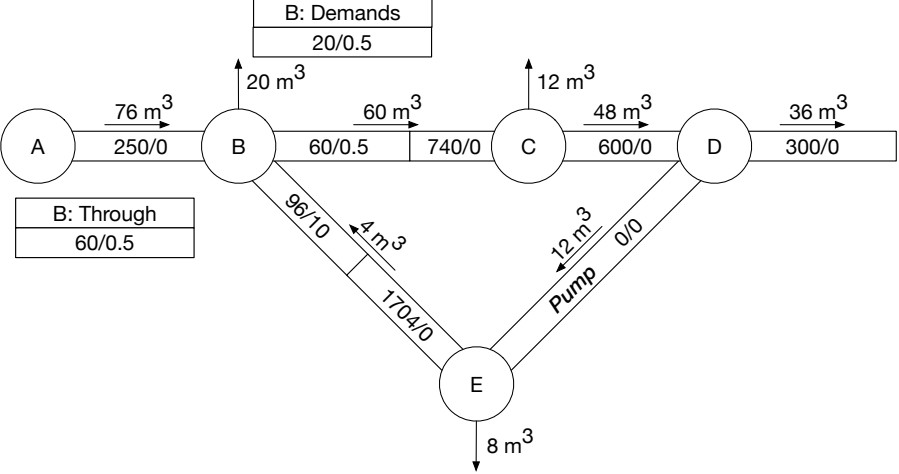

**Figure C5.** Example: Network at the end of the time step.

of 10 mg L⁻¹). Parcels Seg2, Seg3, and Seg4 have the same concentration. The status of the network at the end of the time step is shown in Fig. C5.

In general, in applications using the event-driven algorithm there are multiple interim steps within each water-quality time step. Each of these steps corresponds to an event in the simulation. The preceding discussion mentions the various interim steps included in the time step considered, but emphasizes the method used to address situations involving a recirculating flow. It does not focus explicitly on the the various interim steps themselves. The interim steps in the example are provided explicitly in Table C1, which shows event by event the changes that take place during the time step considered in the example.

The columns in Table C1 correspond to either a link in the network used in the example or to a demand at one of its nodes. The rows correspond to the steps used in the algorithm to route water quality through the network. In this example there are six interim steps between the beginning and end of the water-quality time step. The entries in the table correspond to water



**Table C1.** Water-quality routing for the example using the event-driven approach. Entries in the table describe water parcels, giving their volume and concentration (volume/concentration). Results are shown for one water-quality time step. The last row in the table gives the volume of water moved in a time step.

| Time or volume | Location[a,b] | | | | | | | |
|---|---|---|---|---|---|---|---|---|
| | A-B | $D_B$ | B-C | $D_c$ | C-D | D-E | $D_E$ | E-B |
| Time = 0 | 250/0 | 20/? | 800/0 | 12/? | 600/0 | 0/0 | 8/? | 1700/0, 100/10 |
| Step 1 | 76/0, 250/0 | 20/? | 800/0 | 12/? | 600/0 | 0/0 | 8/? | 1700/0, 100/10 |
| Step 2[c] | 250/0 | 19/0, 1/? | 57/0, 3/?, 800/0 | 12/? | 600/0 | 0/0 | 8/? | 1700/0, 100/10 |
| Step 3 | 250/0 | 19/0, 1/? | 57/0, 3/?, 740/0 | 12/0 | 48/0, 600/0 | 0/0 | 8/? | 1700/0, 100/10 |
| Step 4 | 250/0 | 19/0, 1/? | 57/0, 3/?, 740/0 | 12/0 | 600/0 | 12/0, 0/0 | 8/? | 1700/0, 100/10 |
| Step 5 | 250/0 | 19/0, 1/? | 57/0, 3/?, 740/0 | 12/0 | 600/0 | 0/0 | 8/0 | 4/0, 1700/0, 100/10 |
| Step 6 | 250/0 | 19/0, 1/10 | 57/0, 3/10, 740/0 | 12/0 | 600/0 | 0/0 | 8/0 | 1704/0, 96/10 |
| Time = a time step | 250/0 | 20/0.5 | 60/0.5, 740/0 | 12/0 | 600/0 | 0/0 | 8/0 | 1704/0, 96/10 |
| Vol. moved (m³) | 76 | 20 | 60 | 12 | 48 | 12 | 8 | 4 |

[a] A-B, etc are links. $D_B$, etc are nodal demands.

[b] A question mark (?) means that the concentration has not yet been determined.

[c] The two parcels in $D_B$ correspond to Seg2 in Fig. C2 and the two trailing parcels in B-C correspond to Seg3 in the same figure. Seg1 in the figure corresponds to an intermediate step before the creation of Seg2 and Seg3. It is not associated with any link or demand and therefore is not included in this table.

parcels. The same notation used above to describe a water parcel is used in the table. For example, 250/0 in the first row of the first column indicates that there is a water parcel with a volume of 250 m³ and a concentration of zero in Link A-B at the beginning of the time step. More than one entry indicates that there is more than one water parcel associated with the link or demand at that specific step in the process. The leading parcel in each entry is the rightmost one.

5   In Step 1, a water parcel (76/0) is added to Link A-B due to the inflow of 76 m³ of water to the link from Node A. The concentration of the new parcel is the same (zero) as the concentration of the parcel already in the link (250/0) and the two parcels are combined. Because the combined volume of the parcels now exceeds the volume of the link, the excess volume is moved (Step 2) to Node B, where it is combined with a water parcel from Link E-B (4/?) and then split to accommodate the demand for Node B ($D_B$ in the table) and outflow for the node. As discussed above, the concentration for the parcel arriving

10   from Link E-B has not yet been determined, so incomplete parcels must now be used. Excess water is moved step by step through the network until it is removed by downstream nodes (at Node D) or by demands. By Step 5 the concentration of the inflow from Link E-B can be determined and the incomplete parcels can be completed (Step 6). By the end of the time step the water added at the beginning of the time step has moved through the network and volumes and concentrations of all water parcels and demands have been determined. The conditions at the beginning of the time step correspond to those in Fig. C1





and those at the end of the time step correspond to those in Fig. C5. The conditions in Step 2 correspond to those in Fig. C2 and those in Step 6 correspond to those in Fig. C4.

*Competing interests.* The authors declare that they have no conflict of interest.

*Disclaimer.* This paper has been subjected to EPA's review and has been approved for publication. The views expressed in this paper are those
of the authors and approval does not signify that the contents necessarily reflect the views of the Agency. Mention of trade names, products, or services does not convey official EPA approval, endorsement, or recommendation. Because of the confidentiality of the information, the identity of the real WDSs used in this paper and any information that could be used to identify the systems cannot be disclosed.

The submitted manuscript has been created by UChicago Argonne, LLC, Operator of Argonne National Laboratory ("Argonne"). Argonne, a U.S. Department of Energy Office of Science laboratory, is operated under Contract No. DE-AC02-06CH11357. The U.S. Government
retains for itself, and others acting on its behalf, a paid-up nonexclusive, irrevocable worldwide license in said article to reproduce, prepare derivative works, distribute copies to the public, and perform publicly and display publicly, by or on behalf of the Government. The Department of Energy will provide public access to these results of federally sponsored research in accordance with the DOE Public Access Plan (http://energy.gov/downloads/doe-public-access-plan).

*Acknowledgements.* The U.S. Environmental Protection Agency's (EPA) Office of Research and Development funded, managed, and par-
ticipated in the research described here under an interagency agreement. Work at Argonne National Laboratory was sponsored by the EPA under an interagency agreement through U.S. Department of Energy Contract DE-AC02-06CH11357. All post-simulation data analysis and preparation of graphics for this paper were done using R (R Core Team , 2017). Network plots were produced in R using epanetReader (Eck, 2016). Four internal reviewers provided helpful comments on a draft version of this paper.





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
