# Peer review of "Mass imbalances in EPANET water-quality simulations"

_Drinking Water Engineering and Science, 2017_

## Short Comment (SC1) · 26 Sep 2017

Thank you for your work and effort in this analysis as I believe this is an important topic. However, I did have a few comments on the paper that are more based on using the software for roughly 18 years as an engineering consultant that may be relevant to you. In EPANET and any hydraulic modeling work, the general Rule of thumb requires that the Water Quality timestep and rule timestep be no larger than 1/10th the hydraulic timestep or significant errors can occur. This is why this ratio is stated as the default value for the water quality timestep. The article on page 4 line 13 notes that "Except as noted, all simulations used a hydraulic time step of 3600 s" but the discussion does not seem to recognize that the reason the default WQ timestep is set at a limit of 1/10th the hydraulic timestep is due to accuracy concerns. Your results seem to confirm this

quite clearly that WQ timesteps in excess of 1/10th of the hydraulic timestep can result in significant Mass Balance errors and that this is why this rule of thumb ratio is so critical. If I read Figure 1 properly, the results for 300 s and smaller WQ timestep result in predictions with Mass Balance ratios that appear to mostly fall between 1.0 and 1.1 with an occasional outlier closer to 1.2. This needs to be an essential point to this analysis that EPANET already recommends this key ratio in the WQ timestep in order to get reasonable WQ results as it appears to fail to acknowledge this key point. I would highly recommend that this key ration of hydraulic timestep to WQ timestep be incorporated in the report as it is clearly very critical and essential for good WQ results. The modeler always has to make sure the WQ timestep is always 1/10th (or smaller) than the Hydraulic timestep as the ratio of these two values is often the key factor in reasonable WQ results. Simply shrinking the WQ timestep without any acknowledgement of its ratio to the hydraulic timestep is generally not recommended. The ratio of the two values is often the key to good WQ results. As a hydraulic modeler and engineering consultant of over 18 years I can attest that this ratio is always the most important ratio necessary to get reasonable results in any WQ model. In addition, I recommend that users who are concerned with verifying that the predicted results are as close as possible to actual values that the user test his hydraulic timestep by reducing the hydraulic timestep in half and adjusting his WQ timestep accordingly and comparing results between the two runs until both sets of results match within a reasonable tolerance, as this has been demonstrated in multiple models I have been aware of to verify that the hydraulic results are as accurate as possible, before proceeding to using them for WQ analyses. If the results do not match favorably, then the user should again half the hydraulic timestep and adjust the WQ timestep accordingly and repeat the analysis. If the two sets compare favorably, the user can be confident in using the larger of the two hydraulic timesteps and adjusted WQ timestep as what is needed for the most accurate WQ results for that model. This is essentially critical for models with multiple tanks in close proximity to each other that can create mathematical anomalies in EPANET ( large induced flows between the tanks) when larger hydraulic timesteps

of 1 hour are used. Under those circumstances reducing the hydraulic timestep to between 5 and 15 minutes and reducing the WQ timestep appropriately to 1/10th of the Hydraulic timestep is often sufficient to improve WQ predictions significantly. In light of this, I always recommend anyone looking to get highly accurate WQ concentrations to verify for their model if they need to reduce their hydraulic timestep The only downside to reducing the hydraulic timestep is often that run times for the analysis get longer and longer as the hycraulic timestep is reduced. This can be significant though as WQ simulations are often need to be run for long periods of time in order for them to achieve "steady state" conditions where the results follow a consistent repeating pattern over time.

I would also recommend that the user have a brief explanation of how EPANET calculates Concentrations at junctions (See EPANET 2.0 Help Page 193-199) in regards to discussions of WQ at dead end junctions. Since all flows in the pipes are "numerical calculations" even pipes connected to junctions that are on dead ends "can" have very small flows in them. Due to this, when pipe flows are low that WQ anomalies can occur at junctions connected to dead ends as the mass concentration at the junction is a weighted average based on the flows of all pipes flowing into the junction. This can periodically create short term anomalies in WQ results if the other pipes connected to the junction are also very small as the numerical "flow" in the dead end line can become more significant. This is usually seen as more of an issue in Water age calculations where the dead end junction can have a large water age, but is one reason dead end junctions in WQ analysis are a known issue that can cause short term oddities under certain conditions. Reducing the hydraulic timestep can often assist in this as you note at the bottom of page 12. Lastly in regards to the Conclusions and recommendations, a reduction in the Hydraulic timestep coupled with a WQ timestep 1/10th are often sufficient to improve WQ results. It would be of interest to see how changing the Hydraulic timestep and maintaining the correct ratio of WQ timestep would impact your results and recommendations as well.

---

## Short Comment (SC2) · 26 Sep 2017

The authors have presented both empirical and conceptual evidence to support the claim that EPANET 2.x can exhibit mass imbalance during water quality simulations. For that work the community should be grateful. For my part, the authors will know that the following comments are meant to be constructive; my desire is to help improve the material for more widespread comprehension and collaboration.

The outline of an event-driven algorithm (begin pg. 14/L29), while carefully worded, has a logical inconsistency: (p15/L12) "Nodes are processed in an arbitrary order as long as all inflow paths to a node have water parcels with a known constituent concentration", and then later (p15/L19) "situations can occur in which there are nodes for

which constituent concentrations have not yet been determined for all inflow links. In these cases, an incomplete parcel is created". The second statement would seem to render the first as misleading, or incomplete. Further, is the first statement to be taken literally? That is to say, if all inflow paths to a node do NOT have parcels with known concentration, then would it be true that the nodes are processed in NON-arbitrary order? If this passage is to be an overview of an algorithm, it should be reviewed for logical consistency.

Appendices B-C are extremely illustrative for those readers patient enough to wade through the diagrams and realistic numbers presented (speaking especially of App. B). I would submit that the same concepts communicated in Appendix B could be more concisely framed with a much more pared-down example. A very simple straight-pipe configuration with carefully chosen round-number initial conditions can be made to exhibit the mass imbalance problem, and would have the added benefit of being both intuitively obvious to the hurried, and being "hand-workable" for the more careful reader.

The Recommendations (p20), while well-intentioned, cannot all be supported by the paper's discussion. In particular, the suggestion that "(1) The default water-quality time step [...] should be 60 s" is both jarring and incorrect. The paper delves into great detail about the causes and nature of the mass imbalance phenomenon. It can be clearly understood well before section 6 that mass generation/removal occurs when: 1) the volume carried through a link during a timestep exceeds its geometric volume, and 2) the water quality segments through the transported volume vary spatially.

Both of these conditions are absolutely linked to the particular network being analyzed - in particular the relationship between each link and its volume transported over each timestep. If it is the authors' belief that the current "default" water quality timestep (which may be acceptable for network "A") is worse than a 60-second timestep (which may be not small enough for network "B"), then the recommendation is at worst lacking in nuance. In any case, the guidance of a default 60-second timestep and a do-notexceed of 300-seconds simply cannot be rationalized. Recommendation (1) should be changed to something similar to: - EPANET's current water quality engine should be made to issue a warning/error statement when conditions exist that could lead to mass generation/loss, or when such occurence is detected.

Furthermore, the Recommendations do not coherently describe their urgency or expose any relevant interdependencies. For instance, if (4 - replacement of algorithm) is carried out, then items (1-3) relating to selection of timestep and reporting on mass balance are rendered moot: "The method actually does not require an independent water-quality time step: the simulation is event driven as long as the hydraulic conditions do not change. Because by construction the method accounts for every individual water parcel, its resulting MBR will always be 1.0" (p15/L26). To improve this section, it should be separated into two tiers of recommendations: those that can be accomplished in short order to verify good quality results, and those that can be developed to perfection on a longer timescale.

---

## Short Comment (SC3) · 6 Oct 2017

We appreciate the interest in our work and the comments provided.

To briefly summarize, three comments were provided: (1) The water-quality time step should be less than or equal to one-tenth of the hydraulic time step; (2) The discussion of water quality at dead-end junctions should include an explanation of how EPANET calculates concentrations at junctions; and (3) The paper should examine how changing the hydraulic time step while maintaining the water-quality time step equal to one-tenth of the hydraulic time step affects results and recommendations.

(1) We appreciate the discussion related to best practices for water-quality modeling with EPANET. The comment provides empirical suggestions for improving the results of

water-quality simulations. However, the fundamental issue being raised in the paper is that, in general, the water-quality routing algorithm used in EPANET does not conserve constituent mass. The paper assumes that the hydraulic solution is accurate. The only issue being examined is mass conservation given a hydraulic solution. As the examples in the paper show, a water-quality time step much shorter than one-tenth of the hydraulic time step may be needed to obtain acceptable mass balance and constituent concentrations. Simply decreasing the hydraulic time step and reducing the water quality time step so that it equals one-tenth of the hydraulic time step does not address the fundamental issue, which is the limitations in the water-quality routing algorithm. The purpose of the paper is to demonstrate the limitations in the current algorithm and, hopefully, as a result motivate its replacement with a new algorithm that conserves mass and provides accurate results for constituent concentrations. If an algorithm similar to the even-driven algorithm presented in the paper were available in EPANET, the water-quality time step could be set equal to the hydraulic time step. Selecting a water-quality time step would not then be a significant issue. We do not anticipate a discussion in the paper of best practices for water-quality simulations using the current version of EPANET. If an algorithm similar to the event-driven one presented in the paper were incorporated in EPANET, the issues raised in the comment should disappear.

(2) We will add a parenthetical note on p. 11 at the end of the first sentence on Line 11 stating that in dead-end areas with anomalous flows a potential also exists for anomalous concentrations: "(EPANET determines concentrations of constituents in outflows from a node using the flow weighted sum of inflow concentrations. If other inflows are very small, such anomalous flows could be significant in relative terms and result in concentration anomalies as well.)"

(3) The purpose of the paper is to demonstrate that EPANET does not always conserve mass, explain why this occurs, and show that a different water-quality routing algorithm can eliminate the problem. The results of further analysis using different hydraulic time

steps and a fixed ratio for the water-quality and hydraulic time steps would not affect the conclusions and recommendations of the paper, which are related to the subject of mass conservation. Mass imbalances can occur for networks with very short pipe segments when there is a spatial gradient in concentration. An improved algorithm is needed. See the response to the first comment.
* * *

---

## Short Comment (SC4) · 6 Oct 2017

We appreciate the interest in our work and the comments provided.

To briefly summarize, three comments were provided: (1) There is a logical inconsistency in the description of the event-driven algorithm; (2) The example in Appendix B should be simplified; and (3) The recommendations should be improved.

(1) Thanks for pointing out the inconsistency in the description of the algorithm. We will revise the discussion.

(2) The suggestion to provide a simplified example in Appendix B is reasonable; a simpler example would be easier for the reader to understand. However, accepting this suggestion would require constructing an artificial example. The example used in

[Figure]

Appendix B is based on a real, reproducible situation in Network N1. We believe that the extra complexity is justified because it provides an example that can be reproduced independently using one of the networks from the paper and demonstrates how mass imbalance can occur in a actual case. Opinions obviously can differ on this point.

(3) The comment on the recommendations deserves some discussion. Our goal is to provide recommendations that best serve users of EPANET and thoughts from users are welcomed. The comment has two parts. The first relates to the first recommendation in the paper. The second claims that the recommendations "do not coherently describe their urgency or expose any relevant interdependencies". We will consider the second part of the comment first.

We first provide suggestions for time steps for use "with the current time-driven water-quality algorithm", something that any user can do right away. Therefore, this recommendation could be considered the most urgent. The second recommendation is to add a capability to EPANET to provide reports on mass balance. Some effort is involved (although it is available already in TEVA-SPOT for those who want to use it), but it seems like the next, relatively easy thing to do. (An explicit warning could also be recommended, as suggested.) The third recommendation applies "[w]hen a capability to obtain an evaluation of mass balance is available", clearly relating it to the previous recommendation. The fourth recommendation is to replace the water-quality algorithm with one that conserves mass. It seems obvious that when this is accomplished there will be little motivation for a user to consider the first three recommendations. Perhaps the last recommendation is the most urgent; if it were followed, the problem would be eliminated. Therefore, the order of urgency could be Recommendation 4, followed by "in the meantime" 1, 2, and 3. Or it could be 1, 2, and 3 to first address the needs of current users, followed by, or in parallel with, Recommendation 4. The current list of recommendations does explicitly or implicitly show interdependencies. It also is ordered by urgency (at least according to one possible opinion about urgency).

The first part of the comment says that the recommendation related to specific time

steps is "both jarring and incorrect" and then goes on to say that the first recommendation should be that EPANET should be modified to provide a warning/error statement if problems or potential problems related to mass conservation are identified. Good suggestion. However, what should EPANET users do right now before such warnings are available? Our recommendation is to use a shorter time step. A shorter time step will likely reduce the magnitude of any mass imbalances. The logical alternative to providing such a recommendation seems to be a recommendation that all use of EPANET for water quality simulations be halted until the software can be modified to provide warnings or an accounting of mass balance or until a new water-quality algorithm is available. (Alternatively, we could suggest that all users use TEVA-SPOT. :)) The recommendation does not seem too jarring or incorrect; it provides a workaround until something better is available. Hopefully, users have been alerted to potential problems.

Good comment. Again, we welcome discussion on the subject of recommendations and will consider separating recommendations into two tiers, those that can be carried out soon and those that require a longer time to implement.

---

## Editor Comment (EC1) · R. Shang (Editor) · 19 Oct 2017

I believe that short comments provided by Mr. Patrick Moore and Mr. Sam Hatchett are comprehensive and constructive, and can be considered as full review.

---

## Author Comment (AC1) · 2 Nov 2017

Dear Topical Editor, Please accept our responses to final author comments for the manuscript titled, "Mass imbalances in EPANET water-quality simulations - dwes-2017-28". Here are our responses.

Mass imbalances in EPANET water-quality simulations

Michael J. Davis, Robert Janke, and Thomas N. Taxon

Response to comments We appreciate the efforts of the commenters and the Topical Editor. The comments received are repeated below and the responses provided expand, revise, or finalize our previous responses.

Comments by P. Moore Thank you for your work and effort in this analysis as I believe this is an important topic. However, I did have a few comments on the paper that are more based on using the software for roughly 18 years as an engineering consultant that may be relevant to you.

Comment 1. In EPANET and any hydraulic modeling work, the general Rule of thumb requires that the Water Quality timestep and rule timestep be no larger than 1/10th the hydraulic timestep or significant errors can occur. This is why this ratio is stated as the default value for the water quality timestep. The article on page 4 line 13 notes that "Except as noted, all simulations used a hydraulic time step of 3600 s" but the discussion does not seem to recognize that the reason the default WQ timestep is set at a limit of 1/10th the hydraulic timestep is due to accuracy concerns. Your results seem to confirm this quite clearly that WQ timesteps in excess of 1/10th of the hydraulic timestep can result in significant Mass Balance errors and that this is why this rule of thumb ratio is so critical. If I read Figure 1 properly, the results for 300 s and smaller WQ timestep result in predictions with Mass Balance ratios that appear to mostly fall between 1.0 and 1.1 with an occasional outlier closer to 1.2. This needs to be an essential point to this analysis that EPANET already recommends this key ratio in the WQ timestep in order to get reasonable WQ results as it appears to fail to acknowledge this key point. I would highly recommend that this key [ratio] of hydraulic timestep to WQ timestep be incorporated in the report as it is clearly very critical and essential for good WQ results. The modeler always has to make sure the WQ timestep is always 1/10th (or smaller) than the hydraulic timestep as the ratio of these two values is often the key factor in reasonable WQ results. Simply shrinking the WQ timestep without any acknowledgement of its ratio to the hydraulic timestep is generally not recommended. The ratio of the two values is often the key to good WQ results. As a hydraulic modeler and engineering consultant of over 18 years I can attest that this ratio is always the most important ratio necessary to get reasonable results in any WQ model. In addition, I recommend that users who are concerned with verifying that the predicted results are as close as possible to actual values that the user test his hydraulic timestep by reducing

the hydraulic timestep in half and adjusting his WQ timestep accordingly and comparing results between the two runs until both sets of results match within a reasonable tolerance, as this has been demonstrated in multiple models I have been aware of to verify that the hydraulic results are as accurate as possible, before proceeding to using them for WQ analyses. If the results do not match favorably, then the user should again half the hydraulic timestep and adjust the WQ timestep accordingly and repeat the analysis. If the two sets compare favorably, the user can be confident in using the larger of the two hydraulic timesteps and adjusted WQ timestep as what is needed for the most accurate WQ results for that model. This is essentially critical for models with multiple tanks in close proximity to each other that can create mathematical anomalies in EPANET (large induced flows between the tanks) when larger hydraulic timesteps of 1 hour are used. Under those circumstances reducing the hydraulic timestep to between 5 and 15 minutes and reducing the WQ timestep appropriately to 1/10th of the Hydraulic timestep is often sufficient to improve WQ predictions significantly. In light of this, I always recommend anyone looking to get highly accurate WQ concentrations to verify for their model if they need to reduce their hydraulic timestep The only downside to reducing the hydraulic timestep is often that run times for the analysis get longer and longer as the [hydraulic] timestep is reduced. This can be significant though as WQ simulations are often need to be run for long periods of time in order for them to achieve "steady state" conditions where the results follow a consistent repeating pattern over time.

Response. During water-quality simulations, EPANET should conserve mass. In general, it does not do this. The purpose of the paper is to demonstrate that, in general, it does not conserve mass. We appreciate the comment and recognize that some readers might misinterpret some of our results as being a consequence of not maintaining a ratio of water-quality time step to hydraulic time step of 1/10 or less. We include a water-quality time step of 900 s in our examples because some EPANET users do use this value and because it extends the range of our results and helps show their overall trend. To address this comment and to alert readers of the paper that mass imbalances

are due to a fundamental problem involving the water-quality routing algorithm used in EPANET and not due to poor modeling practices, we will add text at several locations in the paper.

In the Abstract we will add the following text at Lines 5-6 on p. 1:

"This paper provides examples illustrating mass imbalances and explains how such imbalances can occur because of fundamental limitations in the water-quality routing algorithm used in EPANET. In general, these limitations cannot be overcome by the use of improved water-quality modeling practices."

In the Introduction, at the end of the second full paragraph on p. 2, we will add the following text beginning at Line 22:

"The water-quality routing algorithm used in EPANET does not ensure conservation of mass and large imbalances can occur because of fundamental limitations in the algorithm. Good modeling practice requires using a water-quality time step that is less than or equal to one-tenth of the hydraulic time step, which was 3600 s for the example shown in Fig. 1. Although mass imbalances for all injection nodes for Network N1 can be minimized, but not eliminated, by using a water quality time step of 60 s (one-sixtieth of the hydraulic time step), simply reducing the time step will not, in general, ensure mass conservation. In general, mass conservation cannot be ensured with the use of a nonzero water-quality time step."

In the Conclusion, the second sentence will be modified at Line 28 on p. 18 to read as follows:

"Significant mass imbalances can occur when modeling water quality, even for water-quality time steps considerably shorter than those commonly used with EPANET and that are consistent with good modeling practices."

A new paragraph will be added in the Conclusion at Line 13 on p. 18:

"Failure to conserve constituent mass is the result of fundamental limitations in the

water-quality routing algorithm used in EPANET. The algorithm does not ensure mass conservation."

Comment 2. I would also recommend that the user have a brief explanation of how EPANET calculates Concentrations at junctions (See EPANET 2.0 Help Page 193-199) in regards to discussions of WQ at dead end junctions. Since all flows in the pipes are "numerical calculations" even pipes connected to junctions that are on dead ends "can" have very small flows in them. Due to this, when pipe flows are low that WQ anomalies can occur at junctions connected to dead ends as the mass concentration at the junction is a weighted average based on the flows of all pipes flowing into the junction. This can periodically create short term anomalies in WQ results if the other pipes connected to the junction are also very small as the numerical "flow" in the dead end line can become more significant. This is usually seen as more of an issue in Water age calculations where the dead end junction can have a large water age, but is one reason dead end junctions in WQ analysis are a known issue that can cause short term oddities under certain conditions. Reducing the hydraulic timestep can often assist in this as you note at the bottom of page 12.

Response. We will add a parenthetical note on p. 11 at the end of the first sentence on Line 11 stating that in dead-end areas with anomalous flows a potential also exists for anomalous concentrations:

"(EPANET determines concentrations of constituents in outflows from a node using the flow-weighted sum of inflow concentrations. If other inflows are very small, such anomalous flows could be significant in relative terms and result in concentration anomalies as well.)"

Comment 3. Lastly in regards to the Conclusions and recommendations, a reduction in the Hydraulic timestep coupled with a WQ timestep 1/10th are often sufficient to improve WQ results. It would be of interest to see how changing the Hydraulic timestep and maintaining the correct ratio of WQ timestep would impact your results and recom-

mendations as well.

Response. The purpose of the paper is to demonstrate that EPANET does not always conserve mass, explain why this occurs, and show that a different water-quality routing algorithm can eliminate the problem. The results of further analysis using different hydraulic time steps and a fixed ratio for the water-quality and hydraulic time steps would not affect the conclusions and recommendations of the paper, which are related to the subject of mass conservation. Mass imbalances can occur for networks with very short pipe segments when there is a spatial gradient in concentration. The current algorithm cannot adequately handle such situations. An improved algorithm is needed. That is our major recommendation.

COMMENTS by S. Hatchett

The authors have presented both empirical and conceptual evidence to support the claim that EPANET 2.x can exhibit mass imbalance during water quality simulations. For that work the community should be grateful. For my part, the authors will know that the following comments are meant to be constructive; my desire is to help improve the material for more widespread comprehension and collaboration.

Comment 1. The outline of an event-driven algorithm (begin pg. 14/L29), while carefully worded, has a logical inconsistency: (p15/L12) "Nodes are processed in an arbitrary order as long as all inflow paths to a node have water parcels with a known constituent concentration", and then later (p15/L19) "situations can occur in which there are nodes for which constituent concentrations have not yet been determined for all inflow links. In these cases, an incomplete parcel is created". The second statement would seem to render the first as misleading, or incomplete. Further, is the first statement to be taken literally? That is to say, if all inflow paths to a node do NOT have parcels with known concentration, then would it be true that the nodes are processed in NON-arbitrary order? If this passage is to be an overview of an algorithm, it should be reviewed for logical consistency.

Response. The text will be revised to eliminate the inconsistency. The text on Lines 11 to 29 on p. 15 will be replaced with the following (new text is highlighted):

"The event-based, water-quality routing algorithm used here moves homogeneous volumes of water (water parcels with a uniform concentration of a water-quality constituent) through a network. Initially, water parcels are accumulated at all nodes where water enters the system. Nodes with accumulated water parcels from all inflow links are processed in an arbitrary order. Mixing or combining of water parcels occurs at nodes based on the inflow rates of the links flowing into the nodes. Water parcels are combined if the absolute difference between their concentrations is less than some specified amount (the quality tolerance), consistent with the approach used in EPANET 2. After parcels are combined at a node, any nodal demand is removed; the remaining water parcels then are split based on the flow rates of the links flowing from the nodes. These parcels are added to lists of parcels for the downstream links. Any volume in excess of the volume of a link is removed from the leading parcels and placed at the downstream node for further processing. That node is then added to the set of nodes with accumulated water parcels waiting to be processed. Due to recirculating flows, situations can occur in which none of the nodes waiting to be processed has accumulated water parcels on all inflow links. In such cases, an incomplete parcel with the volume that will be moved, but an unspecified concentration, is created for each inflow link that does not have an accumulated inflow. These incomplete parcels are moved, combined, and split in the same manner as parcels for which constituent concentration has been determined; however, internal references are maintained that allow concentrations to be updated when parcels for which concentrations have been determined arrive at a node for which incomplete parcels were created. Flow reversals between hydraulic time steps are accommodated in the same manner as in EPANET 2. The event-driven simulation method provides results that do not depend on the water quality time step if it is equal to or shorter than the hydraulic time step. The method actually does not require an independent water-quality time step: the simulation is event driven as long as the hydraulic conditions do not change. Because by construction the

method accounts for every individual water parcel, its resulting MBR will always be 1.0. An example illustrating the operation of the algorithm using a case with recirculating flow is provided in Appendix C."

Comment 2. Appendices B-C are extremely illustrative for those readers patient enough to wade through the diagrams and realistic numbers presented (speaking especially of App. B). I would submit that the same concepts communicated in Appendix B could be more concisely framed with a much more pared-down example. A very simple straightpipe configuration with carefully chosen round-number initial conditions can be made to exhibit the mass imbalance problem, and would have the added benefit of being both intuitively obvious to the hurried, and being "hand-workable" for the more careful reader.

Response. The suggestion to provide a simplified example in Appendix B is reasonable; a simpler example would be easier for the reader to understand. However, accepting this suggestion would require constructing an artificial example. The example used in Appendix B is based on a real, reproducible situation in Network N1. We believe that the extra complexity is justified because it provides an example that can be reproduced independently using one of the networks from the paper and demonstrates how mass imbalance can occur in an actual case.

Comment 3. The Recommendations (p20), while well-intentioned, cannot all be supported by the paper's discussion. In particular, the suggestion that "(1) The default water-quality time step [...] should be 60 s" is both jarring and incorrect. The paper delves into great detail about the causes and nature of the mass imbalance phenomenon. It can be clearly understood well before section 6 that mass generation/removal occurs when: 1) the volume carried through a link during a timestep exceeds its geometric volume, and 2) the water quality segments through the transported volume vary spatially.

Both of these conditions are absolutely linked to the particular network being analyzed

- in particular the relationship between each link and its volume transported over each timestep. If it is the authors' belief that the current "default" water quality timestep (which may be acceptable for network "A") is worse than a 60-second timestep (which may be not small enough for network "B"), then the recommendation is at worst lacking in nuance. In any case, the guidance of a default 60-second timestep and a do-not exceed of 300-seconds simply cannot be rationalized. Recommendation (1) should be changed to something similar to: - EPANET's current water quality engine should be made to issue a warning/error statement when conditions exist that could lead to mass generation/loss, or when such occurrence is detected.

Furthermore, the Recommendations do not coherently describe their urgency or expose any relevant interdependencies. For instance, if (4 - replacement of algorithm) is carried out, then items (1-3) relating to selection of timestep and reporting on mass balance are rendered moot: "The method actually does not require an independent water-quality time step: the simulation is event driven as long as the hydraulic conditions do not change. Because by construction the method accounts for every individual water parcel, its resulting MBR will always be 1.0" (p15/L26). To improve this section, it should be separated into two tiers of recommendations: those that can be accomplished in short order to verify good quality results, and those that can be developed to perfection on a longer timescale.

Response. Although we disagree with some of the specific statements made in the comment, we accept the overall comment and will revise our recommendations using a tiered approach. The Recommendations section (p. 20) will be revised to read as follows:

"On the basis of results presented here, we recommend that the water-quality algorithm used in EPANET be replaced with one that conserves mass and provides accurate concentration estimates. Until such a change can be accomplished, we recommend the following: 1. Capabilities should be added to EPANET to produce reports on the mass balance of water quality constituents and to provide warning or error statements

when conditions are present that could result in a failure to conserve constituent mass or when such a failure actually occurs. 2. When a capability to obtain an evaluation of mass balance is available, the water-quality time step should be selected so that acceptable mass balances are obtained. 3. As long as a time-driven algorithm is used, some value for a default water-quality time step is needed. To reduce opportunities for mass imbalances to occur, the current default value of 300 s should be reduced."